# Membrane interactions of the globular domain and the hypervariable region of KRAS4b define its unique diffusion behavior

Debanjan Goswami[1†], De Chen[1†], Yue Yang[2], Prabhakar R Gudla[1], John Columbus[1], Karen Worthy[1], Megan Rigby[1], Madeline Wheeler[1], Suman Mukhopadhyay[1], Katie Powell[1], William Burgan[1], Vanessa Wall[1], Dominic Esposito[1], Dhirendra K Simanshu[1], Felice C Lightstone[2], Dwight V Nissley[1], Frank McCormick[3], Thomas Turbyville[1*]

[1]NCI RAS Initiative, Cancer Research Technology Program, Frederick National Laboratory for Cancer Research, Frederick, United States; [2]Biosciences and Biotechnology Division, Lawrence Livermore National Laboratory, Livermore, United States; [3]UCSF Helen Diller Family Comprehensive Cancer Center, School of Medicine, University of California, San Francisco, San Francisco, United States

**Abstract** The RAS proteins are GTP-dependent switches that regulate signaling pathways and are frequently mutated in cancer. RAS proteins concentrate in the plasma membrane via lipid-tethers and hypervariable region side-chain interactions in distinct nano-domains. However, little is known about RAS membrane dynamics and the details of RAS activation of downstream signaling. Here, we characterize RAS in live human and mouse cells using single-molecule-tracking methods and estimate RAS mobility parameters. KRAS4b exhibits confined mobility with three diffusive states distinct from the other RAS isoforms (KRAS4a, NRAS, and HRAS); and although most of the amino acid differences between RAS isoforms lie within the hypervariable region, the additional confinement of KRAS4b is largely determined by the protein's globular domain. To understand the altered mobility of an oncogenic KRAS4b, we used complementary experimental and molecular dynamics simulation approaches to reveal a detailed mechanism.

**\*For correspondence:**
turbyvillet@mail.nih.gov

[†]These authors contributed equally to this work

**Competing interests:** The authors declare that no competing interests exist.

## Introduction

RAS is an oncoprotein that functions as a molecular switch at the apex of a signaling network (*Stephen et al., 2014*) regulating cell differentiation and sustaining cell proliferation, survival, and migration. In oncogenic mutants, this molecular switch mechanism is damaged, and RAS becomes constitutively active—or locked in a GTP-loaded state. In this active state, RAS is in a conformation that favors interaction with several effectors, including RAF, a serine/threonine kinase that, when complexed with RAS at the membrane, phosphorylates its substrate MEK and initiates the mito-genic-activated protein kinase signaling cascade (*Moodie et al., 1993*; *Van Aelst et al., 1993*; *Warne et al., 1993*). RAS biology is characterized by its dynamic association with the plasma membrane. The protein's lateral mobility, nanoclustering of both RAS and lipid species, and interactions between specific lipid species and amino acid residues in the C-terminal hypervariable region (HVR) are all necessary for its function; however, precisely how this is regulated, and in what order events occur has not been fully characterized (*Murakoshi et al., 2004*)((*Lommerse et al., 2005*; *Nan et al., 2015*; *Zhou et al., 2017*).

RAS proteins associate with the plasma membrane (PM) through the HVR. These 22 to 23 amino acids mediate interactions with the lipid bilayer and contain key residues that are post-translationally modified (*Ahearn et al., 2012*). All RAS isoforms contain a C-terminal cysteine residue that is farnesylated and carboxymethylated. HRAS, NRAS, and KRAS4a isoforms contain one or two additional cysteine residues that can be reversibly palmitoylated (*Figure 1a*). In contrast to the other isoforms, KRAS4b is not palmitoylated, but contains a series of six adjacent charged lysine residues (*Figure 1a*). These lysines carry positive charges at a neutral pHand favor electrostatic interactions with anionic lipids, present in cellular membranes enriched with phosphatidylserine and other negatively charged lipids (*Ahearn et al., 2012*). These electrostatic interactions facilitate lipid-based partitioning and sorting of KRAS4b into the disordered domain of membrane (*Zhou et al., 2017*).

The PM is a highly heterogeneous organelle composed of approximately half lipid and half protein by mass (*Cooper, 2000*). The physical and chemical properties of the lipids, proteins, and associated actin cytoskeleton create a multi-tiered, hierarchical 2D structure where molecules diffuse between compartments with length scales on the order of hundreds of nanometers and encounter subdomains with smaller dimensions (*Kusumi et al., 2012*). Diffusing molecules such as RAS traverse these hierarchical, non-equilibrated environments, which create highly transitory, reversible and dynamic subdomains. Using single molecule FRET experiments, Marakoski and colleagues showed that the mobility of membrane-bound KRAS and HRAS was reduced upon growth factor stimulation (*Murakoshi et al., 2004*). This suggests that activated or oncogenic RAS molecules become more confined (*Lommerse et al., 2005*; *Murakoshi et al., 2004*). These investigators have proposed models where oncogenic and activated RAS associate in signaling complexes, perhaps in association with scaffolding molecules and the actin cytoskeleton.

In this study, we use single molecule tracking to detect the mobility of RAS in living cells (*Figure 1—figure supplement 1a*). Using advanced optical microscopy and image analysis techniques combined with engineered cell lines expressing a variety of constructs, we characterize and dissect the relative contributions of the G-domain and the HVR region of KRAS4b to its mobility and compare it to the mobility of other RAS isoforms (*Figure 1a*). We observed that KRAS4b has more confined diffusion in the cell membrane compared to other RAS isoforms, and a unique mobility pattern that is best fit by a three-component model composed of fast, intermediate, and relatively immobile states (*Figure 1b*). By assembling a data set with thousands of trajectories from each cell, we observe transitions between these states; and consequently, we can quantify the probabilities that single molecules of RAS will change states. Intriguingly, we observe that these transitions follow a pattern that is suggestive of a molecular assembly process. We hypothesize that these states are dependent on both GTP-bound G-domain interactions and electrostatic interactions of KRAS4b's unique, highly charged HVR. To develop a molecularly detailed understanding, we use atomistic molecular dynamics simulations to probe the relative contributions of these domains. The results of these simulations are well correlated to our experimental results and provided further testable hypotheses that we explore in cell experiments. We propose a model where on tethering to the membrane via its farnesyl tail, KRAS4b molecules explore the membrane lipid environment in a fast-moving state. While exploring the membrane environment, negatively charged lipids cluster around the lysines in the HVR of KRAS4b forming a nanodomain. When KRAS4b molecules are GTP-loaded within these nanoclusters, KRAS4b can interact with effectors and transition to slower moving states where it can further assemble with effectors and lipids to form a relatively immobile and confined complex competent to signal downstream.

## Results

### KRAS4b diffusion in live cells is characterized by a 3-state hidden Markov model

To measure KRAS4b diffusion in living cells, we used total internal reflection microscopy (TIRF), an electron-multiplying charge-coupled device (EMCCD) camera which supports fast frame acquisition and high sensitivity, and bright organic dyes covalently linked to HaloTagged RAS molecules (for clarity, all molecules studied by single molecule tracking in this set up were HaloTagged, and so the HaloTagged nomenclature will be omitted in the rest of the results section), to visualize single molecules of KRAS4b in the plasma membrane at a 10 ms frame rate. Bright and photostable dyes

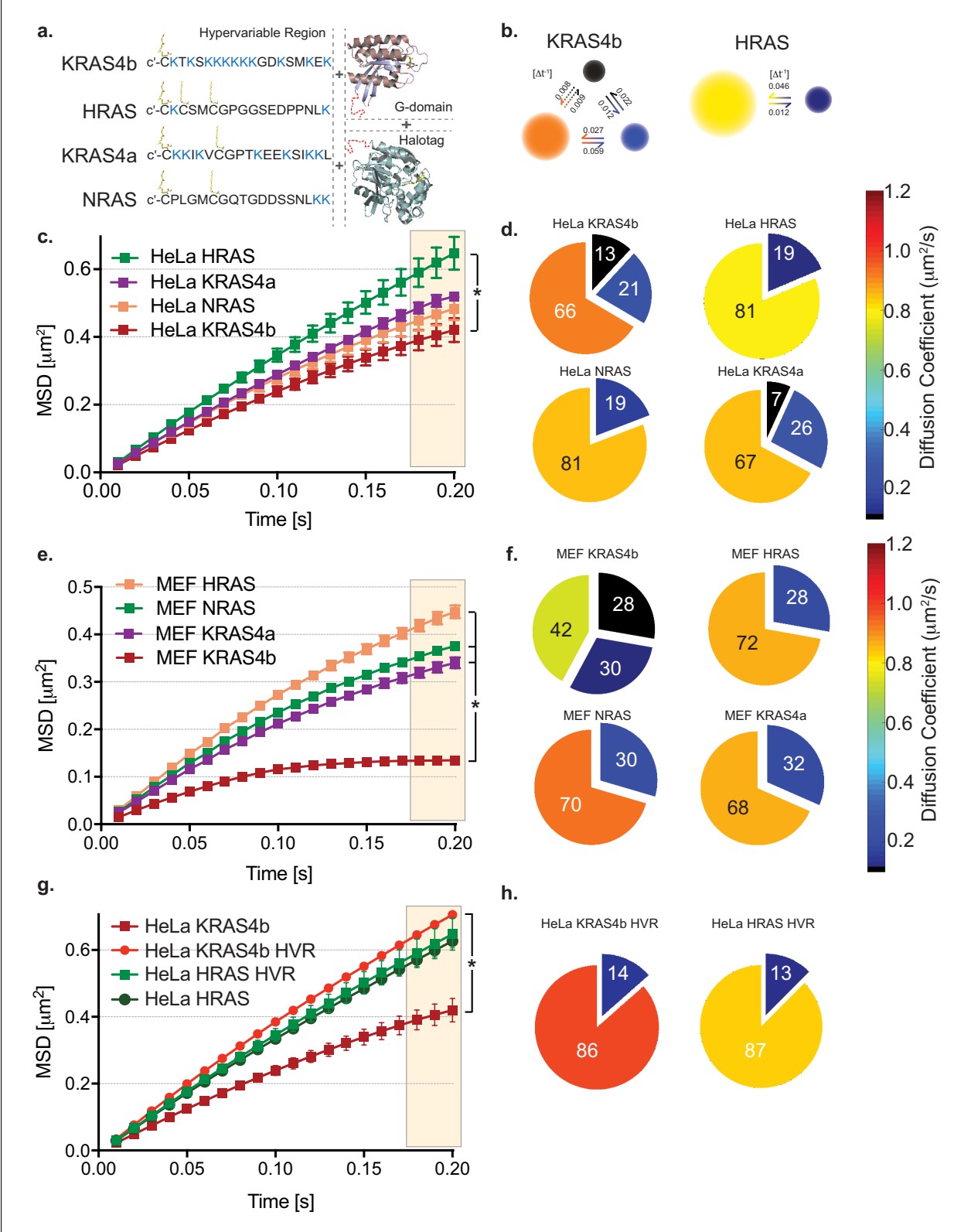

**Figure 1.** HMM and MSD analysis results of KRAS4b and isoforms in HeLa and MEF cells. (a) Illustration of combinations of fusion proteins between hypervariable region (HVR) and G-domain of RAS isoforms with HaloTag that were expressed in HeLa and RAS-less MEF cells. (b) Hidden Markov modeling (HMM) using vbSPT of single molecule tracking (SMT) measurements showed different mobile patterns and was described with three and two diffusive states for KRAS4b and HRAS, respectively. Each diffusion (state) coefficient is pseudo-color coded, as marked by the rainbow scale bar and

*Figure 1 continued on next page*

*Figure 1 continued*

probability of transitions between states per frame rate ($\Delta t$ = 10 ms). (c) The Mean Squared Displacement (MSD) vs. time plot showed highest confinement of KRAS4b whereas the least for HRAS (* indicates three displacement values under shaded area are significantly different, p<0.05). (d) Diffusion coefficients and occupancy (HMM analysis) from various isoforms of RAS molecules on HeLa cell membrane are summarized in pie charts. (e) MSD analysis on SMT measurements on RAS-less MEF cells for various RAS isoforms (* indicates three displacement values under shaded area are significantly different, p<0.05). (f) The summary of diffusive states from HMM analysis on MEF cells are presented in pie charts for KRAS4b, HRAS, KRAS4a and NRAS. (g) MSD plot showing a significant difference (p<0.05) in confinement (bending of curve) of diffusion between KRAS4b full-length and its truncated HVR, whereas no significant difference was found in case of HRAS. (h) Pie charts show both KRAS4b and HRAS HVRs diffusion were best described by a two-state model using HMM analysis.

The online version of this article includes the following video, source data, and figure supplement(s) for figure 1:

**Source data 1.** MSD values over time (plotted in *Figure 1c*) for Halotag-KRAS4b, -KRAS4a, -HRAS, and -NRAS transiently expressed in HeLa cells and recorded with TIRF microscopy.

**Source data 2.** Diffusion coefficients and occupancy fractions obtained by HMM analysis (plotted in *Figure 1d*) of Halotag-KRAS4b, -KRAS4a, -HRAS, and -NRAS transiently expressed in HeLa cells.

**Source data 3.** MSD values over time (plotted in *Figure 1e*) for Halotag-KRAS4b, -KRAS4a, -HRAS, and -NRAS expressed in isogenic Mouse Embryonic Fibroblast cell pools.

**Source data 4.** Diffusion coefficients and occupancy fractions obtained by HMM analysis (plotted in *Figure 1f*) of Halotag-KRAS4b, -KRAS4a, -HRAS, and -NRAS expressed in isogenic Mouse Embryonic Fibroblasts.

**Source data 5.** MSD values over time (plotted in *Figure 1g*) for Halotag-KRAS4b, Halotag-KRAS4b HVR (lacking the G domain), Halotag-HRAS, and Halotag-HRAS HVR transiently expressed in HeLa cells.

**Source data 6.** Diffusion coefficients and occupancy fractions obtained by HMM analysis (plotted in *Figure 1h*) of Halotag-KRAS4b HVR and Halotag-HRAS HVR transiently expressed in HeLa cells.

**Figure supplement 1.** Detection of HaloTagged RAS protein constructs in live cells by single molecule tracking and confocal imaging.

**Figure supplement 2.** HMM and MSD analysis results of KRAS4b G12D diffusion in cancer cell lines.

**Figure supplement 2—source data 1.** MSD values over time (plotted in *Figure 1—figure supplement 2a*) for overexpressed, exogenous Halotag-KRAS4b G12D in a panel of pancreatic cancer cell lines with existing KRAS4b G12D mutations (SU.86.86, hTERT-HPNE , and PANC-1).

**Figure supplement 2—source data 2.** Diffusion coefficients and occupancy fractions obtained by HMM analysis (plotted in *Figure 1—figure supplement 2b*) for overexpressed, exogenous Halotag-KRAS4b G12D in a panel of pancreatic cancer cell lines with existing KRAS4b G12D mutations (SU.86.86, hTERT-HPNE , and Panc-1).

**Figure supplement 3.** Single detection, trajectories of KRAS4b diffusion and Dox-induced KRAS4b diffusion.

**Figure supplement 3—source data 1.** Diffusion coefficients and occupancy fractions obtained by HMM analysis (plotted in *Figure 1—figure supplement 3d*) of Halotag-KRAS4b for increasing concentrations of doxycycline in a dox-inducible Halotag-KRAS4b HeLa cell pool.

**Figure supplement 4.** Basal signaling profiles and Ras expression levels of isogenic MEF pools.

**Figure 1—video 1.** TIRF video microscopy of HaloTag-KRAS4b in a live HeLa cell.

https://elifesciences.org/articles/47654#fig1video1

**Figure 1—video 2.** 16μm x16μm region of interest of the plasma membrane in Hela cell and tracks.

https://elifesciences.org/articles/47654#fig1video2

**Figure 1—video 3.** TIRF video microscopy of HaloTag-KRAS4b-C185S mutant in live Hela cell.

https://elifesciences.org/articles/47654#fig1video3

---

allowed us to observe single molecules of RAS for up to seconds at a time. We used the Variational Bayes for Single Particle Tracking (vbSPT) algorithm (*Persson et al., 2013*), based on a hidden Markov model (HMM), to analyze the heterogeneous mobility of RAS molecules in the plasma membrane, and we identified a three-state model for KRAS4b diffusion in live cell membranes. We observed this behavior in HeLa cells, three different KRAS mutant cancer cell lines (*Figure 1—figure supplement 2*), and mouse embryonic fibroblast (MEF) cells that only express one isoform of human RAS at a time (*Tables 1–2*) In all cells tested, KRAS4b molecules show three different diffusion components. To test whether this three-component system and associated diffusion rates were dependent on RAS density, we developed a DOX-inducible KARS4b cell line to be able to control expression levels. Increasing the density of KRAS4b at the membrane did not appreciably change the diffusion rates or impact the number of states (*Figure 1—figure supplement 3*) (*Table 3*). Representative results in a HeLa cell line include one dominant fast mobile component (0.95+ /- 0.03 $\mu m^2/s$), an intermediate component (0.24+ /- 0.03 $\mu m^2/s$) and a slow component (0.06+ /- 0.01 $\mu m^2/s$), and fractional occupancy of the three states at 64% (+ /- 2), 22% (+ /- 2) and 14% (+ /- 3) respectively (*Table 1*). Mobility of all RAS isoforms and mutants in various cell lines are shown in *Tables 1–7*.

**Table 1.** Diffusion rates and percent occupancy in RAS isoforms, and their HVRs.

| Dataset HeLa | Diffusion coefficient ($\mu m^2$/s) | | | Occupancy (%) | | |
|---|---|---|---|---|---|---|
| | D1 | D2 | D3 | F1 | F2 | F3 |
| KRAS4b | 0.95 ± 0.03 | 0.24 ± 0.03 | 0.06 ± 0.01 | 64 ± 2 | 22 ± 2 | 14 ± 3 |
| KRAS4a | 0.82 ± 0.04 | 0.26 ± 0.01 | 0.05 ± 0.01 | 67 ± 2 | 26 ± 3 | 7 ± 2 |
| NRAS | 0.84 ± 0.04 | 0.23 ± 0.08 | | 81 ± 2 | 19 ± 1 | |
| HRAS | 0.81 ± 0.02 | 0.1 ± 0.03 | | 81 ± 1 | 19 ± 1 | |
| KRAS4b HVR | 0.97 ± 0.03 | 0.1 ± 0.05 | | 87 ± 3 | 13 ± 3 | |
| HRAS HVR | 0.82 ± 0.06 | 0.1 ± 0.03 | | 87 ± 2 | 13 ± 2 | |
| NRAS HVR | 0.95 ± 0.08 | 0.15 ± 0.05 | | 86 ± 1 | 14 ± 1 | |

HMM analysis also returns the transition rate between states (**Persson et al., 2013**), which reflects how frequently KRAS4b switches between its subdiffusive states under the influence of membrane dynamics. The transition probability [$\Delta t^{-1}$] between states for KRAS4b revealed that a direct conversion from a fast diffusing state to the slowest state, or *vice versa*, is over 10 times less likely than *via* the intermediate state (**Figure 1b**). Since KRAS4b molecules must traverse an intermediate state, it implies that KRAS4b diffusion and the mobility changes it undergoes are part of an ordered process on the PM, although whether this represents an assembly or oligomerization process or whether KRAS4b itself is modifying the lipid environment is not known. Interestingly, in comparison to KRAS4b, the fraction of slow moving KRAS4a and NRAS molecules decreased markedly in cells overexpressing these isoforms and becomes lower for HRAS. For HRAS and NRAS we used a two-state diffusion model (**Figures 1b, d and f**) since the occupancy of the slowest diffusing species in the three-state models was less than 5%, and we believe this to be negligible background signal. Taken together, this data suggests that KRAS4b has unique diffusion characteristics in cells.

To evaluate whether the tagged RAS molecules are indeed functional we took advantage of the RAS-dependent MEF system. These cells are dependent on MAP kinase signaling for proliferation, and it is only through the re-introduction of an isoform of RAS, or through the introduction of a constitutively active form of proteins in the MAP kinase pathway, such as BRAF V600E, that the cells will re-enter the cell division cycle (**Drosten et al., 2010**). This is an ideal system for evaluating the ability of specific proteins to restore the MAP kinase signaling required for their proliferation. To verify that the RAS molecules measured on the cytoplasmic face of the plasma membrane in living cells are functionally competent, we integrated HaloTag-RAS fusion constructs in the genomic DNA of MEFs that are devoid of endogenous KRAS, HRAS, and NRAS molecules by viral transduction (**Figure 1— figure supplement 4**). After transduction, we observed that the MEFs were rescued from their quiescence state and reentered the proliferation cycle (**Drosten et al., 2010**). Using imaging, we observed, furthermore, that the tagged KRAS4b molecules can localize correctly to the plasma membrane. Finally, we found that the levels of pAKT, pMEK and pERK are at levels comparable to

**Table 2.** Diffusion rates and percent occupancy of KRAS4b in cancer cell lines

| Dataset KRAS4b | Diffusion coefficient ($\mu m^2$/s) | | | Occupancy (%) | | |
|---|---|---|---|---|---|---|
| | D1 | D2 | D3 | F1 | F2 | F3 |
| HeLa (KRAS4b WT) | 0.95 ± 0.03 | 0.24 ± 0.03 | 0.06 ± 0.01 | 64 ± 2 | 22 ± 2 | 14 ± 3 |
| MEF (KRAS4b WT) | 0.73 ± 0.12 | 0.25 ± 0.09 | 0.05 ± 0.01 | 42 ± 9 | 30 ± 8 | 28 ± 2 |
| PANC-1 (KRAS4b G12D) | 0.84 ± 0.06 | 0.22 ± 0.01 | 0.04 ± 0.01 | 43 ± 6 | 40 ± 4 | 17 ± 2 |
| SU.86.86 (KRAS4b G12D) | 0.8 ± 0.16 | 0.18 ± 0.09 | 0.02 ± 0.01 | 66 ± 14 | 28 ± 9 | 6 ± 5 |
| hTERT-HPNE (KRAS4b G12D) | 0.92 ± 0.06 | 0.25 ± 0.02 | 0.06 ± 0.01 | 55 ± 3 | 29 ± 1 | 16 ± 3 |

**Table 3.** Diffusion rates and percent occupancy in full length KRAS4b with increasing concentrations of doxycycline in a dox-inducible HeLa cell pool

| Dataset HeLa | Diffusion coefficient ($\mu m^2/s$) | | | Occupancy (%) | | |
|---|---|---|---|---|---|---|
| | D1 | D2 | D3 | F1 | F2 | F3 |
| DOX 1 ng/mL | 0.96 ± 0.04 | 0.33 ± 0.04 | 0.09 ± 0.01 | 71 ± 1 | 20 ± 2 | 9 ± 1 |
| DOX 2 ng/mL | 0.93 ± 0.02 | 0.26 ± 0.01 | 0.07 ± 0.002 | 68 ± 2 | 22 ± 2 | 10 ± 1 |
| DOX 5 ng/mL | 0.95 ± 0.04 | 0.27 ± 0.01 | 0.07 ± 0.01 | 66 ± 3 | 24 ± 1 | 10 ± 2 |

WT MEF cells as measured in western blots (*Figure 1—figure supplement 3*). Taken together, these results indicate that the HaloTag fusion proteins are functional, properly localized, and sufficient to restore the appropriate signaling in the cells.

To further confirm that the overexpression of the fusion constructs is not causing any artifactual membrane association, we introduced a point mutation at the terminal cysteine residue to serine (C185S) that would stop the post-translational farnesylation of KRAS4b required for membrane association. This modification prevents stable membrane association in HeLa cells, as observed by a significant reduction in residence time at the membrane (*Figure 1—figure supplement 2c*). We recorded transient flashes of KRAS4b C185S molecules on the PM as opposed to longer, trackable trajectories in the wildtype protein (*Figure 1—figure supplement 3c*).

## KRAS4b diffusion shows anomalous, confined diffusion by Mean Squared Displacement analysis

To further evaluate the unique diffusion properties of KRAS4b, we used Mean Squared Displacement (MSD) analysis to characterize the diffusion behaviors of RAS molecules in the plasma membrane (*Matysik and Kraut, 2014*). In MSD plots, the curvature of the graph indicates the extent of confinement of the molecules as they diffuse in the plasma membrane—the more bent the curve, the greater the confinement. We analyzed each isoform of RAS molecule to see whether diffusion behavior reflects the heterogeneity in RAS membrane association and subdomain preferences, as we might predict based on the differential post-translational lipid modifications across all these isoforms (*Figure 1a*). In the line graphs of the four different RAS species, KRAS4b, KRAS4a, NRAS and HRAS, in HeLa and isoform-specific MEF cells, respectively, we see clear differences in the diffusion behavior of KRAS4b compared to the other isoforms (*Figure 1c–f*). Our analysis (*Figure 1c and e*) shows that KRAS4b has the highest level of confinement (anomalous diffusion), whereas HRAS was the least confined. Interestingly, KRAS4a and NRAS, both with one palmitoylation site, are intermediate to HRAS and KRAS4b (*Figure 1c and e*). To assess the generalizability of KRAS4b diffusion behavior, we overexpressed and evaluated KRAS4bG12D mobility in various cell lines (*Figure 1—figure supplement 2*), such as pancreatic cancer lines harboring endogenous mutant KRAS4bG12D, a ductal epithelial cell line, and importantly, in MEF cell lines that are devoid of any endogenous RAS protein (*Figure 1e*), and in all cases, we observed similar diffusion behavior.

**Table 4.** Diffusion rates and percent occupancy in full length KRAS4b wildtype and with G-domain mutations ESR, GNK, and HEK

| Dataset HeLa | Diffusion coefficient ($\mu m^2/s$) | | | Occupancy (%) | | |
|---|---|---|---|---|---|---|
| | D1 | D2 | D3 | F1 | F2 | F3 |
| KRAS4b | 0.84 ± 0.02 | 0.22 ± 0.01 | 0.05 ± 0.003 | 57 ± 1 | 27 ± 1 | 14 ± 1 |
| KRAS4b ESR (D126E/T127S/K128R) | 0.91 ± 0.04 | 0.25 ± 0.02 | 0.06 ± 0.01 | 63 ± 4 | 25 ± 2 | 12 ± 2 |
| KRAS4b GNK (E91G/H94N/H95K) | 0.85 ± 0.01 | 0.22 ± 0.01 | 0.05 ± 0.01 | 66 ± 1 | 22 ± 2 | 12 ± 1 |
| KRAS4b HEK (Q131H/D132E/R135K) | 0.90 ± 0.06 | 0.24 ± 0.01 | 0.06 ± 0.003 | 58 ± 5 | 28 ± 1 | 14 ± 4 |

**Table 5.** Diffusion rates and percent occupancy in full length KRAS4b, the HVR, and HVR mutants in HeLa cells

| Dataset HeLa | Diffusion coefficient ($\mu m^2$/s) | | | Occupancy (%) | | |
|---|---|---|---|---|---|---|
| | D1 | D2 | D3 | F1 | F2 | F3 |
| KRAS4b | 0.95 ± 0.06 | 0.24 ± 0.03 | 0.06 ± 0.01 | 66 ± 2 | 23 ± 2 | 14 ± 3 |
| 4bHVR | 0.98 ± 0.03 | 0.17 ± 0.03 | | 87 ± 5 | 13 ± 5 | |
| 4bHVR-3A | 1.08 ± 0.06 | 0.13 ± 0.04 | | 87 ± 6 | 13 ± 5 | |
| 4bHVR-5A | 1.07 ± 0.04 | 0.15 ± 0.02 | | 82 ± 8 | 18 ± 6 | |
| 4bHVR-5Ea | 1.21 ± 0.05 | 0.15 ± 0.05 | | 87 ± 3 | 13 ± 4 | |
| 4bHVR-5Eb | N/A | N/A | N/A | N/A | N/A | N/A |

## The higher confinement in mobility is exclusively experienced by full length KRAS4b

Recruitment and association of RAS molecules with the PM requires the HVR, and it is in this region of the protein that the amino acid sequence differs between isoforms (*Figure 1a*). To explore the contribution of the HVR to the increased confinement level of KRAS4b, we compared the MSD behaviors of the full-length proteins with the truncated HVRs with the entire G-domain deleted (*Figure 1g*, *Figure 1—figure supplement 1b*). We found that the HaloTagged HVRs express well, localize to the plasma membrane, and laterally diffuse at the membrane forming trajectories which we can track. In comparison with KRAS4b, all HVRs show very little confinement (*Figure 1g*). Notably, we determined that the KRAS4b HVR, when expressed as a truncation mutant without the G-domain, shows free mobility in contrast to the full-length protein, which shows confinement. This data suggests that the confined behavior of KRAS4b is dependent on the G-domain of the protein and is not attributable to the electrostatic and other side chain interactions with the PM contributed by the HVR.

## KRAS4b confinement may in part be due to dynamic G-domain contacts with the lipid bilayer

To better characterize the possible interactions of the G-domain of KRAS4b with lipids, we conducted atomistic simulations of KRAS4b molecules in association with a lipid bilayer (POPC: POPS, 80:20). These simulations showed that G-domain residues formed transient interactions with lipid head groups and we modeled these residues onto a structure of KRAS4b (*Figure 2a*) selecting a series of mutants in three clusters that also correspond to amino acid differences between RAS isoforms. We tested these mutants in simulations and in living cells. Atomistic simulation of the various KRAS4b's G-domain mutants on membrane (POPC: POPS, 80:20) altered the distribution of positions of the molecule (*Figure 2b*). Consistent with the results from the simulation, MSD analysis of single molecule trajectories in living cells showed that the 4b-GNK mutant (N-like: E91G/H94N/H95K) and the 4b-ESR mutant (H-like: D126E/T127S/K128R) had less confinement (*Figure 2c*), and vbSPT analysis showed that the fast-moving fraction of the 4b-ESR mutant increased, and its slow-moving components decreased (*Figure 2d*) (*Table 4*).

**Table 6.** Diffusion rates and percent occupancy in full length KRAS4b, KRAS4b Q61R, KRAS4b Y40C, and KRAS4b Y40C-Q61R in HeLa cells

| Dataset HeLa | Diffusion coefficient ($\mu m^2$/s) | | | Occupancy (%) | | |
|---|---|---|---|---|---|---|
| | D1 | D2 | D3 | F1 | F2 | F3 |
| KRAS4b Q61R | 0.90 ± 0.07 | 0.25 ± 0.02 | 0.06 ± 0.01 | 51 ± 5 | 33 ± 3 | 16 ± 1 |
| KRAS4b | 0.88 ± 0.06 | 0.25 ± 0.02 | 0.08 ± 0.01 | 56 ± 2 | 28 ± 8 | 16 ± 1 |
| KRAS4b Y40C | 0.94 ± 0.07 | 0.27 ± 0.02 | 0.07 ± 0.01 | 66 ± 6 | 23 ± 5 | 12 ± 2 |
| KRAS4b Y40C-Q61R | 0.92 ± 0.05 | 0.31 ± 0.04 | 0.08 ± 0.02 | 62 ± 6 | 26 ± 4 | 11 ± 2 |

**Table 7.** Diffusion rates and percent occupancy in Ras isoforms, and MEF serum starved cell recovery with serum complete media

| Dataset KRAS4b | Diffusion coefficient (μm²/s) | | | Occupancy (%) | | |
|---|---|---|---|---|---|---|
| | D1 | D2 | D3 | F1 | F2 | F3 |
| MEF | 0.73 ± 0.12 | 0.25 ± 0.09 | 0.05 ± 0.01 | 42 ± 3 | 30 ± 3 | 28 ± 4 |
| MEF (srm 0') | 0.72 ± 0.02 | 0.22 ± 0.02 | 0.07 ± 0.01 | 50 ± 3 | 27 ± 0.8 | 23 ± 3 |
| MEF (srm 15') | 0.77 ± 0.04 | 0.22 ± 0.02 | 0.06 ± 0.01 | 47 ± 3 | 28 ± 3 | 25 ± 2 |
| MEF (srm 60') | 0.85 ± 0.03 | 0.21 ± 0.02 | 0.06 ± 0.01 | 44 ± 4 | 29 ± 3 | 27 ± 2 |

## The charge state of 4bHVR only influences the fast state: fast component dominated by charge of HVR

In addition to the insertion of the farnesyl group, the electrostatic interaction between the positively charged HVR and the negatively charged lipid head groups is a necessary second signal for KRAS4b molecules to associate with the membrane (*Ahearn et al., 2012*), and is involved with signaling through both lipid recruitment and effector interactions (*Terrell et al., 2019*; *Zhou et al., 2017*). We modified the charge states of the KRAS4b HVR and tested their mobility following the charge adjustment by transient expression in HeLa cells of HaloTagged fusion proteins lacking the G-domain (*Figure 3a*). For all the charge mutants tested, except for 4bHVR-5Eb, we acquired appreciable single molecule trajectories (*Figure 3b*), indicating that not all electrostatic interactions are necessary for stable membrane association and mobility. We also observed that all the isolated 4bHVRs had a negligible fraction of the slow or immobile component; and therefore, to simplify analysis, we use a two-state model to describe their motion.

*Figure 3c* and (*Table 5*) show the mobility of the two components (a fast state and an intermediate state), as a function of the HVR charge state and demonstrates that the fast component is a function of the lysine residues in the HVRs. We see a linear and significant increase in the fast component diffusion rate as the positive charge density on the HVR is reduced (*Figure 3a* gives the sequence and charge states of the mutated HVRs), and the charge reversal mutant 4bHVR-5EA has the highest mobility in the fast state (~1.2 μm²/s). We do not see a significant difference in the state occupancy of these mutants (*Figure 3d*). In addition to an increase in its fast component diffusion rate, 4bHVR-5Ea shows a shorter residence time on the membrane (*Figure 3e*), suggesting impeded membrane association. The 5Ea mutant also exhibits a lower probability of transitioning from the fast to slow diffusion states (*Figure 3f*). The wild type 4bHVR (WT HVR) construct is fully able to associate with the membrane and has the lowest mobility (corresponding to the fast component of the full-length protein). Interestingly, the HVR charge states did not impact the slower mobility component, which suggests that any diffusing molecule in the membrane can experience local environments of confinement—perhaps due to molecular crowding or structural features in the membrane.

4bHVR-5EB has the same charge distribution as the 4bHVR-5EA but does not associate appreciably with the membrane (*Figure 3—figure supplement 1*), indicating that HVR lysines are not simply bearing charge, but that the position of the charge impacts membrane association. Since 4bHVR-5EA and 4bHVR-5EB have the same total negative charge (−1.1 CC), we attribute their differential membrane association to lysine 184 next to the C terminal (KTKC) and hypothesize that this lysine and the farnesyl group together may be key for KRAS membrane association. In fact, in Dharmaiah et al, the cocrystal structure of KRAS4b and its chaperone PDEδ, shows that K184 forms multiple hydrogen bonds with the main chain in PDEδ, stabilizing the interaction with the chaperone. Consistent with our results, reversal of the charge on K184 would be expected to disrupt this interaction and could disrupt KRAS4b transport to the membrane (*Dharmaiah et al., 2016*). The alanine mutants 4bHVR-5A and 4bHVR-3A retain net positive charges (3.9 and 5.9 CC, respectively) and can associate with the membrane but also with higher mobility than the wild type HVR (8.9 CC).

Since we show that the mobility of the fast component is sensitive to the charge carried by the lysine residues in the HVR, we hypothesized that the free diffusion of RAS in the plasma membrane is driven by electrostatic interactions with negatively charged phospholipids, such as phosphatidylserine. We tested this supposition with molecular dynamics simulations. *Figure 4a* shows a series of snapshots from the atomistic MD simulation (using a CHARMM36 Force Field) of the various HVR mutants and the WT HVR in association with a simple membrane (POPC: POPS, 80:20). The snapshot

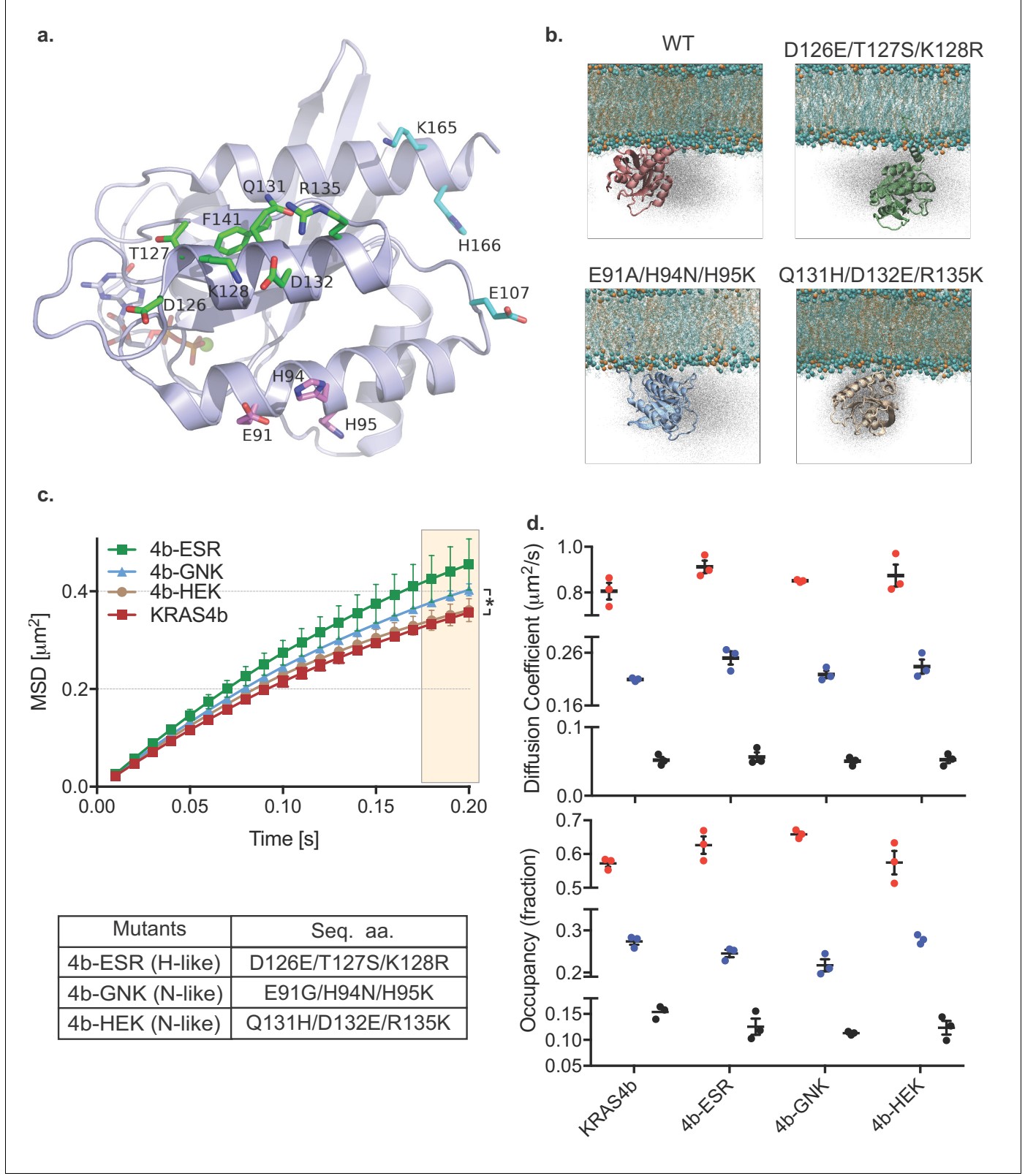

**Figure 2.** HMM and MSD analysis results of KRAS4b with mutated contact points of G-domain with the lipids and the MD simulation of KRAS4b and the same mutant. (a) Ribbon diagram of the G-domain of KRAS depicts locations of transitory contact points with the lipids and the following table shows mutations made to the G-domain to mimic H-like or N-like RAS at those key residues. (b) Snapshots from simulations of KRAS4b on lipid membrane (POPC: POPS, 80:20) are shown where G-domain makes transitory contacts with the lipids *via* salt bridges, with the ghost/shadow

*Figure 2 continued on next page*

*Figure 2 continued*

representation exhibiting space that G-domain has sampled. (**c**) MSD plot from the H-like or N-like mutants are relatively less confined compared to WT-KRAS4b (the asterisk * indicates three displacement values under shaded area are significantly different, p<0.05). (**d**) Diffusion coefficients and occupancy obtained from HMM analysis are compared for each mutant; no significant difference was found between mutants.

The online version of this article includes the following source data for figure 2:

**Source data 1.** MSD values over time (plotted in *Figure 2c*) of Halotag-KRAS4b and G-domain mutants 4b-ESR (HRAS-like), 4b-GNK (NRAS-like), and 4b-HEK (NRAS-like).

**Source data 2.** Diffusion coefficients and occupancy fractions obtained by HMM analysis of Halotag-KRAS4b and G-domain mutants 4b-ESR (HRAS-like), 4b-GNK (NRAS-like), and 4b-HEK (NRAS-like).

---

represents the average positioning data and shows that the WT HVR is the most closely associated with the membrane—due to electrostatic interactions of the lysines with negatively charged lipids. *Figure 4b* shows a similar group of snapshots, this time with a systematic increase (Δ10% steps) in the relative proportion of phosphatidylserine (POPS) in the simulation (from 0% to 40%). In these snapshots, we see that the average position of the WT HVR in the cloud of positions sampled becomes more closely associated with the membrane as the PS concentration increases (*Figure 4b*). The average distance of the lipid head group and amino acid side chains is quantified in line plots and shows the same trend (*Figure 4c and d*). Furthermore, the simulation result shows that 4bHVR-5EB does not interact with the membrane, which agrees well with our experimental result showing that the construct forms few trajectories (*Figure 3b*), and poor membrane localization (*Figure 3—figure supplement 1*). A snapshot of POPS clustering around the HVR is shown (*Figure 4e*), and quantification of the ratio of POPS:POPC during the evolution of the simulation through time (*Figure 4f*) shows that WT HVR coalesces and reorganizes the negatively charged lipids compared to the HVR mutants with the neutral alanine side chains replacing lysines. Together these data show the critical function of the electrostatic interactions between the positively charged lysine residues in the HVR and the negatively charged head groups in the membrane and show that the HVR is dynamically clustering PS in the membrane leading to membrane reorganization.

## Mutations in the G-domain do not change lateral mobility; however, activated KRAS shows higher occupancy in the intermediate and slow states

We used KRAS4b and its mutants Q61R and Y40C (*Figure 5a*) to analyze KRAS mobility and relate mobility to biology and signaling. The Q61R oncogenic mutant has intrinsically poor GTP hydrolysis activity and is therefore constitutively loaded with GTP. We expect that it will be a highly active isoform interacting with effector molecules such as RAF (*Novelli et al., 2018*). The Switch 1 Y40C mutation has been shown previously to disrupt RAF binding (*Joneson et al., 1996*). An MSD versus time plot (*Figure 5b*) shows the levels of confinement for the different species when transiently expressed in HeLa cells. Consistent with a model that confinement is in part due to assembly with effectors, we found that the activated Q61R RAS molecule is the most confined in mobility, while the Q61R/Y40C double mutant is the least confined in mobility among the three species. *Figure 5c* shows the mobility of the four types of KRAS molecules with their respective state occupancy fractions in *Figure 5d*. The mobility of all components of the three species are similar in magnitude. The Q61R mutant has a significantly higher fraction in the slow-moving component, and a smaller fraction in the fast-moving component compared to the Y40 and Q61R/Y40C double mutant. In contrast, Q61R/Y40C, a switch one mutant impaired in its ability to bind its effector RAF, shows a significantly higher fast component fraction and smaller slower component fractions (*Table 6*). These findings indicate that the confinement level of KRAS4b is related to the activity of RAS; specifically, GTP-loaded oncogenic KRAS4b is in a conformation that favors interaction with effectors, and therefore these molecules are more likely to be found in signaling complexes. We infer that the increase state occupancy of Q61R mutants in the slow state indicates that KRAS4b signaling complexes are relatively immobile. Conversely, Y40C mutants are not competent to interact with effectors, especially RAF1 (*Joneson et al., 1996*), and we see fewer Y40C molecules in the slow state.

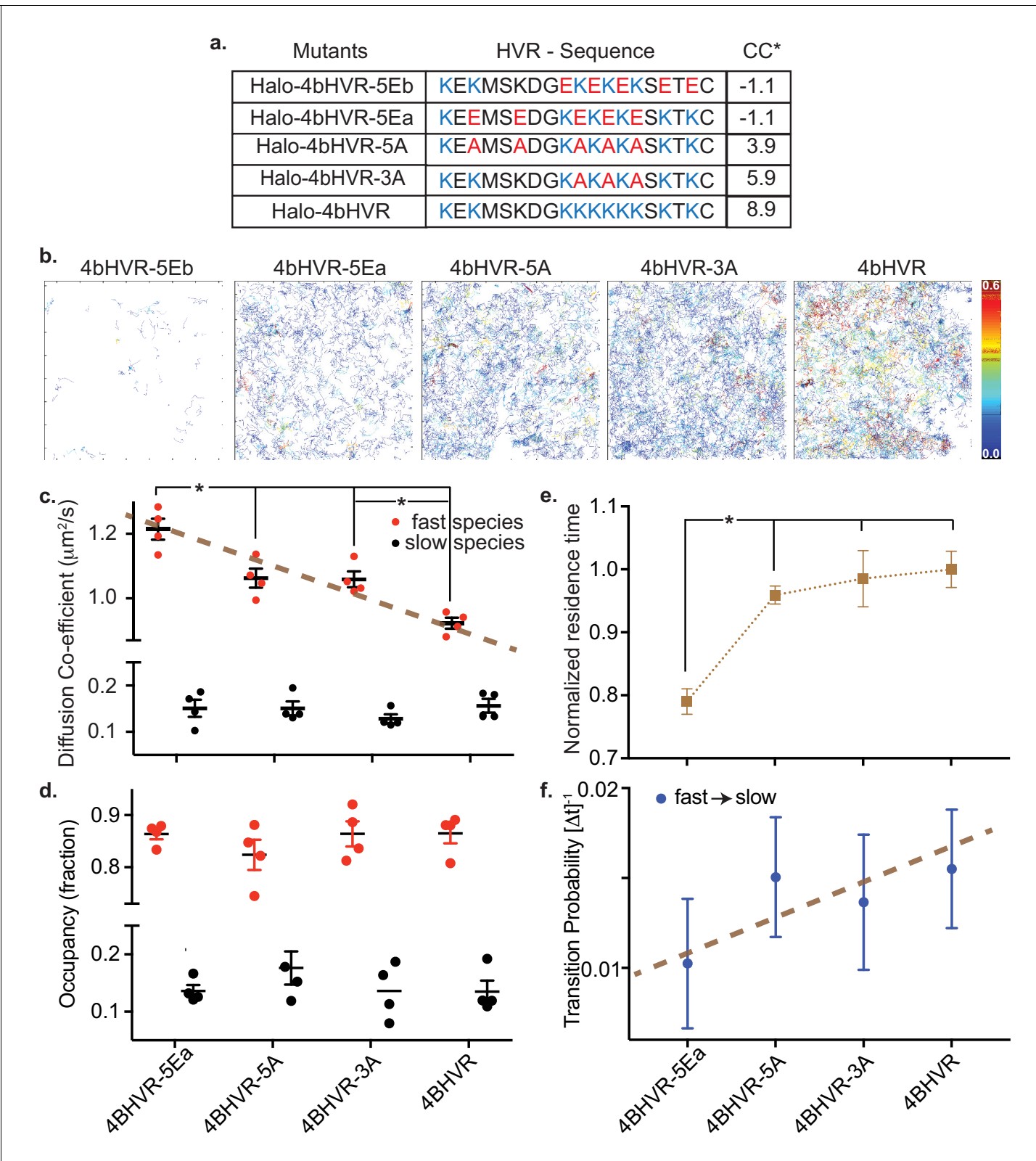

**Figure 3.** HMM analysis of KRAS4b HVR and mutations that adjusted the charges of the HVR. (a) Table shows HaloTag HVR (of KRAS4b) constructs over-expressed in HeLa cells by transient transfection with charge-neutral and charge-reversed substitution mutations of lysine residues with varying net charge content (*CC). (b) Single molecule tracks are shown for WT HVR and each of the mutant HVRs, color-coded according to their residence time at the membrane. The color bar encodes time from 0.0 s to 0.6 s. (c and d) Diffusion coefficient and occupancy of the fast and slow diffusing species are

*Figure 3 continued on next page*

*Figure 3 continued*

plotted in red and black solid circles respectively for each charge-altered mutant. While the fast diffusing species exhibit a gradual, significant ($p<0.05$) increase in diffusion coefficient as the lysine charges are neutralized and then reversed, the slow species remain the same and the relative fraction of fast and slow diffusing species remain unchanged. (**e**) Normalized average residence time from more than 5000 tracks for each mutant is shown in the plot. Significant reduction ($p<0.05$) in residence time indicates impaired association between charge-reversed HVR and membrane. (**f**) Graph shows transition probabilities from fast to slow diffusive state for each charge-altered mutant.

The online version of this article includes the following source data and figure supplement(s) for figure 3:

**Source data 1.** Diffusion coefficients and occupancy fractions obtained by HMM analysis (plotted in *Figure 3c and d*) of Halotag-KRAS4b HVR and the charge reversal mutants 5Ea, 5A, and 3A transiently overexpressed in HeLa cells.
**Figure supplement 1.** Localization of KRAS4b HVR mutant constructs.

## Signaling from G-domain influences KRAS4b confinement

To further test the hypothesis that confinement is related to the biological and functional activity of RAS, we serum starved confluent MEF cells (expressing HaloTag-KRAS4b) for 18 hr and performed single molecule tracking experiments. Significantly, MSD analysis shows that RAS molecules in serum starved cells are less confined (*Figure 6a*). When starved cells are rescued with complete medium, the molecules show increased confinement within 15 min. HMM analysis of the same experiment showed a significant increase in fast moving molecules (8%), and a corresponding decrease (5%) in the slow diffusing species (*Figure 6b and c*) (*Table 7*). Taken together with the data with the Q61R and Y40C mutants, these data demonstrate that RAS diffusion behavior correlates with known RAS biochemistry in which activated, GTP-loaded RAS associates with effectors and activates downstream signaling (*Figure 6—figure supplement 1*).

## Discussion

RAS is primarily associated with the inner leaflet of the cell plasma membrane and membrane association is critical to its activity (*Simanshu et al., 2017*). One distinct feature of all RAS isoforms lies in the sequence specificity of their HVRs, which may influence where in the plasma membrane RAS molecules associate or may itself influence the assembly of lipid domains through the recruitment of lipids into nanoclusters. RAS shows a high degree of lateral mobility in the PM, and the PM is heterogenous and contains distinct lipid domains that may influence RAS and cell signaling (*Kusumi et al., 2012*). Understanding how these dynamics are associated with RAS activity, and whether RAS plays a role in regulating these interactions, is a key focus of this work. In this study, we tested the hypothesis that the unique diffusion characteristics of KRAS4b are due to features of the full-length protein, and that these features lead to unique interactions with the PM and to an assembly process of the signaling complex.

KRAS4b shows mobility characterized by three states: a fast, intermediate and slow diffusion rate. The transition probabilities show that the transition between slow and fast states is rare; specifically, the transitions occur through the intermediate state. In light of these findings, we propose a model (*Figure 6d*) in which the fast state represents RAS molecules that are not complexed with effector molecules and are diffusing with lipid molecules, and in which the two slower states represent different points in the assembly process—the intermediate state representing slowing, partially complexed, RAS molecules, and the slow state representing fully assembled RAS molecules in a signaling complex (that might be highly constrained by unique structural features in the plasma membrane such as a actin corral as proposed by *Kusumi et al., 2012*). In support of our model, we can control the fractional occupancy of these states by either serum starving KRAS4b-dependent MEFS, which increases the fast state occupancy, or stimulating starved cells with serum, which increases the number of molecules transitioning to the intermediate and slow states. This along with the mutational analysis we collected in HeLa cells, is strong evidence that transition to the slower states of KRAS4b is dependent on G-domain interactions.

We used simulations of KRAS4b and the 4bHVR in defined lipid environments to observe in atomistic detail the correlation of amino acid and lipid interactions and distance measurements and compared them to our experimental data. Computer simulations predict that the positive charges on the HVR recruit negatively charged lipids and bring the HVR into closer proximity to lipids in the membrane. Correspondingly, in experiments if we neutralize or reverse the positive charges in the HVR,

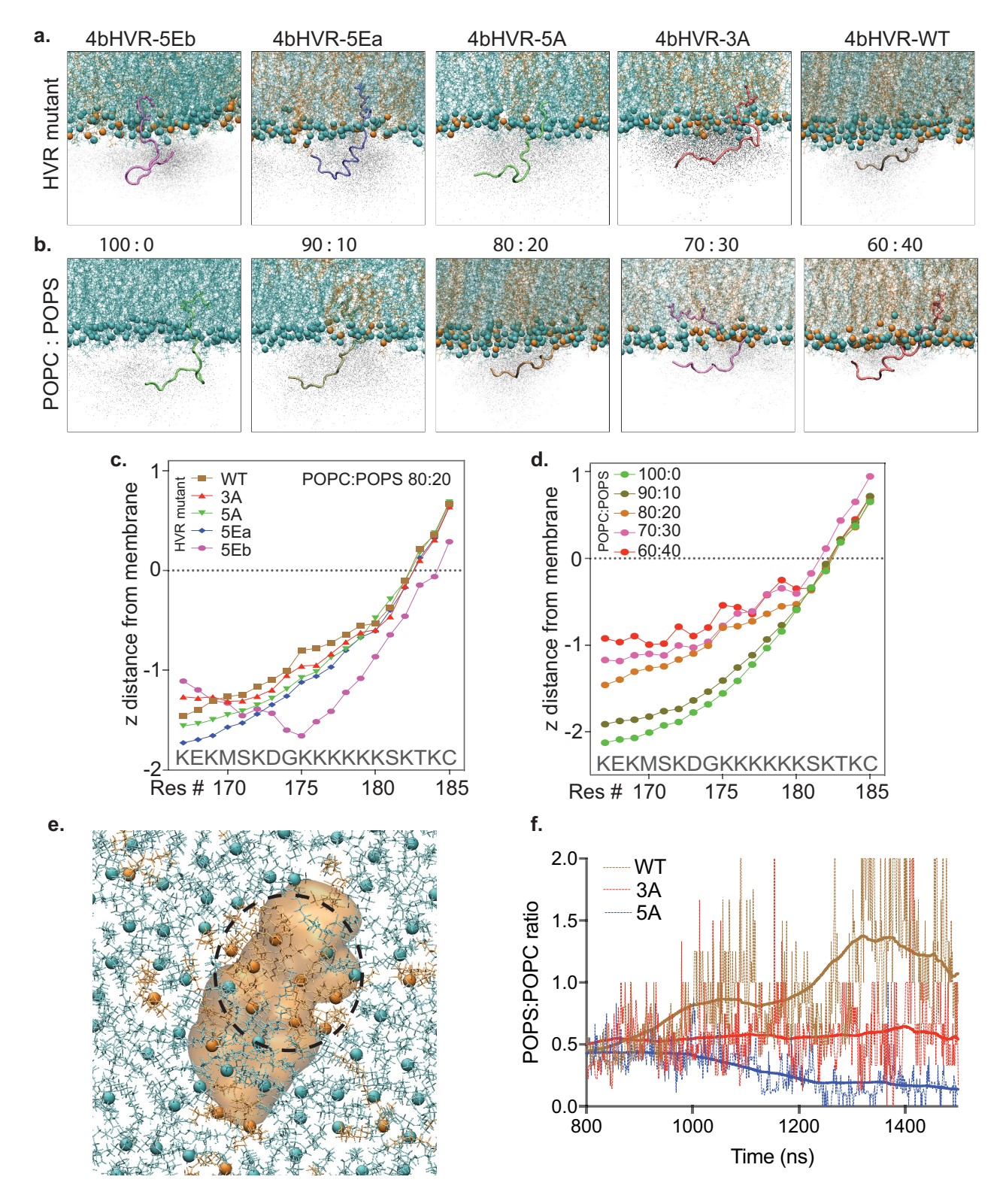

**Figure 4.** Atomistic simulation of KRAS4b HVR and its mutants interacting with the artificial membrane. (**a**) Atomistic simulations from various KRAS4b's HVR mutants on membrane (POPC: POPS, 80:20) are shown in solid representation where the snapshot most closely represents the average positioning data. In addition, the space HVR has sampled are shown in ghost/shadow representation. (**b**) Systematic change of POPS from 0% to 40% in the membrane shows increasing proximity of the HVR to the membrane. (**c and d**) Line plot shows quantification of 'z' distance between lipid head groups

*Figure 4 continued on next page*

*Figure 4 continued*

and individual amino acids (carbon alpha) in the HVR obtained from the simulations. (**e**) A snapshot from the simulation showing clustering of POPS around the HVR because of electrostatic interaction between positively charged lysine residues and anionic lipid head groups. Blue spheres represent POPC, brown spheres represent POPS, and the 4bHVR is modeled in brown (POPC (80%) – POPS (20%)). (**f**) Quantification of POPS: POPC ratio in the proximity of the HVR during the time course of simulation. It shows 4bHVR's ability to concentrate POPS, compared to the charge-deficient mutants, 4bHVR-3A and 4bHVR-5A.

we see increasing diffusion rates in the fast component, suggesting that the recruitment of negatively charged lipids is responsible for the fast component's diffusion rate ($0.97 \ \mu m^2/s \pm 0.03$). Although we do not directly observe the recruitment of negatively charged lipid molecules around the HVR in experimental data (as this is beyond our imaging capabilities), the atomistic MD simulations clearly show this to be the case.

Furthermore, the MSD data gives direct information about KRAS4b mobility and allows us to characterize the qualitative features of RAS diffusion; specifically, the extent to which the molecules show diffusion patterns of free, anomalous, or more constrained diffusion. Using MSD analysis (without any modeling assumptions), we show that KRAS4b shows distinctively confined diffusion behavior compared to the other isoforms or truncation mutants. For membrane-associated molecules, such as lipids and proteins, anomalous diffusion is thought to be caused by molecular crowding, or confinement to specific subdomains in the lipid bilayer. However, we demonstrate that the confinement of KRAS4b is dependent on the G-domain, since the truncated mutant HaloTag-4bHVR shows almost free diffusion. We suspected that growth factor driven signaling followed by molecular assembly among effector and KRAS4b, and other proteins might be responsible for slowing down the molecular diffusion. Indeed, serum-starved MEF cells expressing the full-length protein, show reduced confinement. When we rescue cells with complete, serum containing media, the KRAS4b molecules change their diffusion behavior and become more confined, as indicated by the bend in the MSD plots (*Figure 6a*). Furthermore, oncogenic KRAS4b-Q61R, which is constitutively loaded with GTP, shows more confined behavior than KRAS4b molecules that express a mutation in the switch one binding domain (Y40C), which abrogates RAF engagement with RAS. Therefore, effector binding, and signaling in the cells plays a role in confinement of KRAS4b molecules. Finally, we extended the study using atomistic molecular simulations where we show that specific residues in the G-domain form transient contacts with lipids in the membrane. In follow-up biological experiments, we mutated these residues in KRAS4b to be HRAS-like and NRAS-like, and we saw less confined diffusion for these RAS molecules.

It is important to note that diffusion in the membrane is not linearly dependent on the size of the diffusing species (*Saffman and Delbrück, 1975*); however, these findings suggest that it is both protein-protein and lipid-G-domain interactions that are influencing KRAS4b mobility in ways that are distinct from other isoforms of RAS. We propose in our model that the positive charges on the 4bHVR recruit and cluster negatively charged lipids around RAS molecules (mutation of the HVR lysines leads to a higher diffusion rate, suggesting that these lipid associations contribute to the fast diffusion component), and subsequent G-domain interactions with both effectors and lipids lead to an ordered, stepwise assembly process where KRAS4b molecules are increasingly confined to smaller nanodomains in the formation of a signaling complex (*Figure 6c*).

Our study clarifies several unresolved issues in the field. *Murakoshi et al. (2004)* used a single molecule fluorescence energy transfer (FRET) technique to observe single molecule activation of RAS molecules in the plasma membrane of cells. Using a fluorescently labeled GTP analogue which they microinjected into cells and exogenously expressed fluorescently tagged RAS fusion proteins, they observed single molecule trajectories of activated RAS molecules loaded with the fluorescent GTP molecule and reported slow diffusion rates for the active RAS molecules. *Lommerse et al. (2005)* focused their studies on HRAS. They compared wild type, constitutively active (G12V) and inactive (S17N) forms of HRAS and found that both mutants had two populations of either slow or fast diffusing molecules. In the case of the G12V mutant, they found that the slow-moving molecules were confined to small 200 nm domains, and that wild type proteins were confined to domains of the same size upon insulin stimulation. Consistent with our observations, both studies found that these trajectories were suppressed or immobilized upon activation of the RAS molecule. Although these studies elegantly demonstrate that activation of RAS molecules with GTP loading or oncogenic

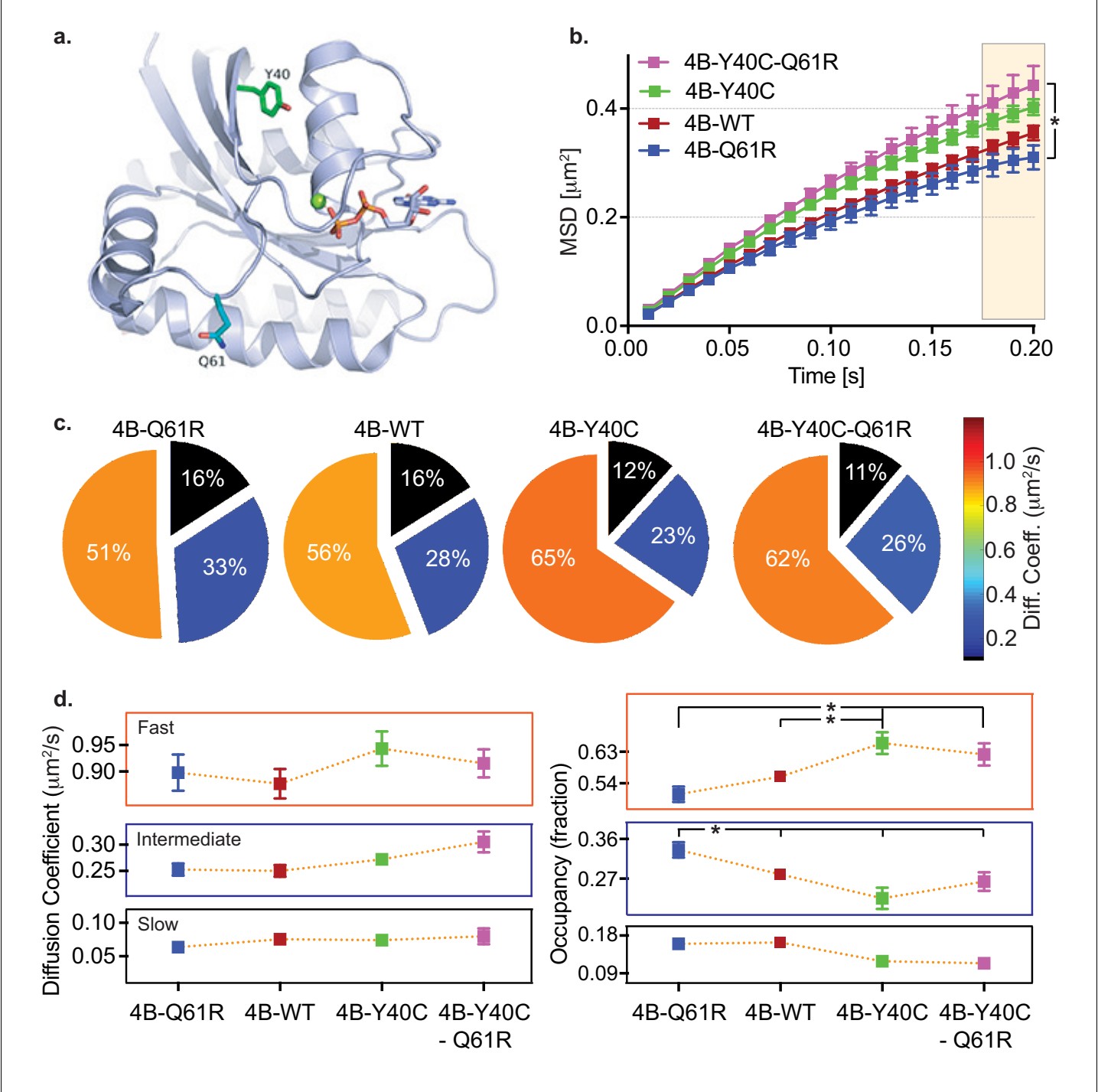

**Figure 5.** HMM and MSD analysis results of KRAS4b and the oncogenic and RAF binding disabled mutants. (a) Structural illustration of KRAS4b in ribbon diagram depicts amino acid locations for the mutations Y40C (green), Q61R (blue), and GTP (yellow) (b) MSD plot showed significant difference in diffusion properties between various KRAS4b mutants transiently expressed in HeLa cells. Increased linearity in MSD profile for RAF binding-deficient mutants Y40C and Q61R/Y40C suggests that the effector interaction is another major contributor in confinement of diffusion of KRAS4b. The three displacement values under shaded area between Q61R and Q61R-Y40C mutants are significantly different, p<0.05. (c and d) Pie charts and line plots show the Y40C and Q61R/Y40C mutants have higher occupancy in the fast-diffusing state while lower occupancy the intermediate-diffusing state as determined by vbSPT HMM analysis. Occupancy of fast and intermediate diffusing species is significantly (p<0.05) different between Q61R/WT and Y40C/Q61R-Y40C variants.

The online version of this article includes the following source data for figure 5:

**Source data 1.** MSD values over time, plotted in *Figure 5b*, of Halotag-KRAS4b, oncogenic KRAS4b-Q61R, Raf-binding deficient mutant KRAS4b Y40C, and the combination mutant KRAS4b-Y40C-Q61R transiently expressed in HeLa cells.

**Source data 2.** Diffusion coefficients and occupancy fractions obtained by HMM analysis (plotted in *Figure 5c*) for Halotag-KRAS4b, -KRAS4b Q61R, -KRAS4b Y40C, and -KRAS4b Y40C-Q61R.

activation changes diffusion properties, we were able to extend these observations and show differences between isoforms and provide further insight to identify both G-domain and HVR contributions to this mobility pattern.

*Zhou et al. (2017)* showed that the amino acid sequence of the HVR of KRAS and the prenyl group function as a code for determining the nature of RAS associated nanoclusters. Using a combination of electron microscopy, spatial mapping, and atomistic simulations, their study suggests that HVR interactions are not just electrostatic in nature, but that specific amino acid side chains direct which types of lipids will cluster in the plasma membrane, and that this may have implications in downstream signaling events. Our studies complement these observations by revealing the dynamics of KRAS4b and that mutations in HVR lysines change the diffusion behavior, indicating that the assembly of signaling complexes may be dependent on the nature of the lipid clusters forming around the HVR side chains. We add an additional level of detail, showing by atomistic molecular dynamic simulations that G-domain residues are forming transitory contacts with lipid, and that mutation in these residues affect diffusion behavior. Interestingly, when we convert these G-domain residues of KRAS4b to mimic HRAS and NRAS, the diffusion properties of the molecule show less KRAS4b-like confined behavior.

In summary, we show that the diffusion of KRAS4b in the PM is best explained by a three-state model in which a slow (immobile) state involves G-domain interactions with both the PM and effector engagement. Although we did not focus our efforts on characterizing them, the other RAS isoforms clearly undergo different mechanisms of dynamic interaction with the plasma membrane, as indicated by both their relative lack of confinement (MSD) and fewer states (vbSPT). Our findings strongly suggest that the KRAS4b-plasma membrane interaction plays an important role in regulating RAS signaling through a stepwise and ordered process that gives rise to a specific transition path with different diffusion properties. Furthermore, our study has implications for understanding the interaction of proteins with the plasma membrane, and the active processes involved in assembly of a signaling complex that involves both protein-lipid and protein-protein interactions.

## Materials and methods

### Protein constructs

HaloTag fusion proteins of RAS family members and hypervariable regions (HVRs) were generated using combinatorial Multisite Gateway (*Wall et al., 2014*). Briefly, three components were mixed in a Multisite LR reaction: a strong CMV51 promoter (att4-att5), a HaloTag (Promega) fusion protein with upstream Kozak initiation sequence and lacking a stop codon (att5-att1), and a standard Gateway Entry clone of the various downstream fusion partners (att1-att2). Correct recombinants were isolated and verified by restriction digest, and transfection-ready DNA was prepared using Qiagen plasmid preparation kits.

Entry clones for some constructs were from the RAS mutant entry clone collection (Addgene); those containing HVR sequences were synthesized directly as Entry clones (ATUM Bio, Inc). The remaining clones were generated by site-directed mutagenesis from RAS mutant entry clone constructs using the Quickchange kit (Agilent). All Entry clones were fully sequence verified prior to subcloning into the final fusion vectors.

### Generation of HaloTagged RAS mutant mouse embryonic fibroblasts

DU1473, a HRAS-/-NRAS-/-, KRASlox/lox, RERT[ert/ert] primary mouse embryonic fibroblast cell line, generated and characterized previously (*Drosten et al., 2010*) was generously provided by Dr. Mariano Barbacid (Spanish National Cancer Research Center (CNIO)). Cells were cultured in Dulbecco's Minimum Essential media containing high glucose and L-glutamine (ThermoFisher, Waltham, MA) supplemented with 10% Fetal Bovine Serum (GE Healthcare, Pittsburgh, PA). Cells were rendered RAS-less by culturing in complete media supplemented with 600 nM 4-hydroxytamoxifen

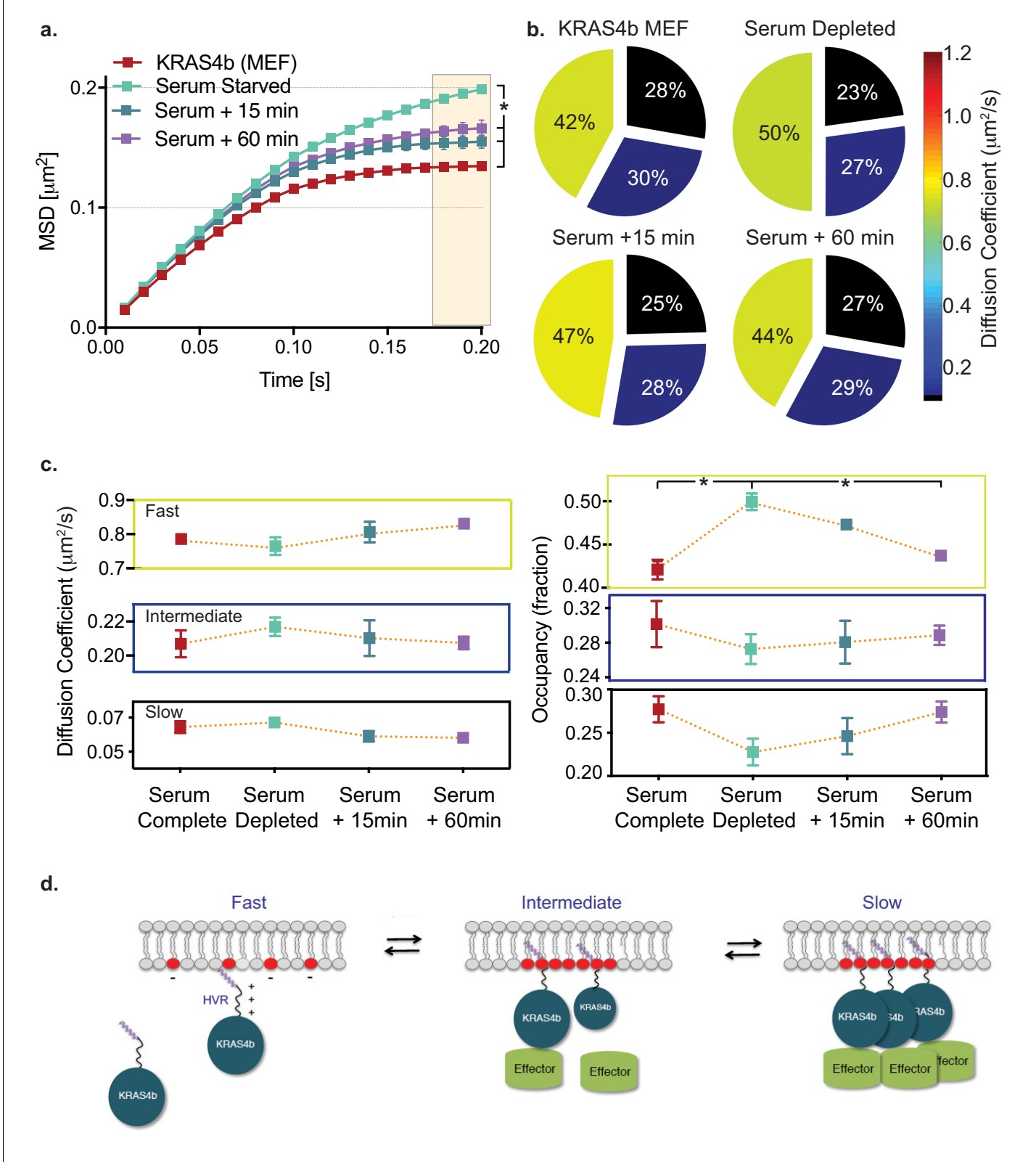

**Figure 6.** HMM and MSD analysis results of KRAS4b diffusion in MEF in starved and serum complete media rescued conditions. (a) MSD plot shows relative changes of confinement in diffusion profile of KRAS4b in MEF cells upon serum deprivation (incubated in 0.1% FBS media for 18 hr). MEF cells that are expressing only the KRAS4b isoform, were deprived from serum for 18 hr and then, rescued with 10% FBS containing DMEM media for 15 mins and 60 mins respectively on each coverslip. The three displacement values under shaded area between serum substituted and depleted are

*Figure 6 continued on next page*

*Figure 6 continued*

significantly different (p<0.05), as indicated by the asterisk '*'. (**b**) vbSPT HMM analysis from same diffusion tracks is displayed on the right-side panel, showing reduced slow-diffusing fraction in serum-deprived state. (**c**) Line plots of KRAS4b diffusion coefficients and state occupancy in complete media, depleted media and rescued with complete media after 15 and 60 min as determined by vbSPT HMM analysis. Occupancy of fast diffusing species is significantly (p<0.05) different between serum substituted and depleted conditions. (**d**) Cartoon model showing KRAS4b dynamics in the membrane. The online version of this article includes the following source data and figure supplement(s) for figure 6:

**Source data 1.** MSD values over time (plotted in *Figure 6a*) of Halotag-KRAS4b in complete serum conditions (10% FBS), serum starved (0.1% FBS 18 hr), and after 15 or 60 mins of rescue with 10% FBS serum.

**Source data 2.** Diffusion coefficients and occupancy fractions obtained by HMM analysis (visualized in *Figure 6b* and *Figure 6c*) for Halotag-KRAS4b in complete serum conditions (10% FBS), serum starved (0.1% FBS 18 hr), and after 15 or 60 min of rescue with 10% FBS serum.

**Figure supplement 1.** Western blot analysis of KRAS4b signaling under serum starvation and recovery with serum complete media.

(Sigma-Aldrich, St. Louis, MO) for 14 days to activate translocation of the estrogen receptor (ER)-fused Cre to the nucleus for removal of the endogenous (floxed) *Kras* genes resulting in G1 arrest. Cell proliferation was 'rescued' by transduction with lentivirus produced by the RAS Reagent Core of the RAS Initiative containing the RAS mutant of interest tagged with HaloTag. Briefly, tamoxifen-treated cells were seeded into six-well plates and incubated overnight at a density that yielded 60–70% confluency at the time of transduction. Media was removed from the cultures and replaced with 1 mL media containing 8 µg/mL hexadimethrine bromide (Sigma-Aldrich, St. Louis, MO). One well from each plate was harvested and counted to calculate the amount of virus to add per well for each transduction. Lentivirus volume corresponding to multiplicities of infection (MOI) of 5 or 10 was added to the hexadimethrine bromide containing media. After 24 hr incubation at 37°C, 5% $CO_2$, the virus containing media was removed and replaced with pre-warmed complete media. Cells were incubated for an additional 24 hr before the media was replaced with pre-warmed selection media containing either Puromycin 2.5 µg/mL or Blasticidin 4 µg/mL. Cell line pools were expanded in selection media for a minimum of 1 week prior to further testing. Cell line pools were verified via PCR to be free of *Mycoplasma* contamination and were Sanger sequenced to verify the insertion of the desired transgene sequence.

## Generation of Dox-inducible HaloTag KRAS4b HeLa cell pool

HeLa cells obtained from ATCC were transduced at MOI 0.05 with lentivirus containing plasmid construct R980-M38-658 (CMV13p>TetOn3G); both plasmid and virus were generated at the RAS Initiative. Cells were cultured using the same DMEM media and 10% Fetal Bovine Serum as described above. After 24 hr, media containing lentivirus was removed and replaced with pre-warmed complete media. After an additional 24 hr, media was removed and replaced with pre-warmed complete media containing 4 µg/mL Blasticidin. Cells were grown under selection for 2 weeks before viable freeze aliquots were stored in vapor phase liquid nitrogen. The transduced HeLa cells were then restarted in culture and transduced at MOI 10 with lentivirus containing plasmid construct R713-M15-663 (TRE3Gp > Halotag7 Hs.KRAS4b) and were subjected to an additional 2 weeks of selection with 4 µg/mL Blasticidin and 1 µg/mL Puromycin prior to use.

## Cell culture, transfection and labeling of HaloTag-Ras

For the experiments with over-expression system, HeLa cells were transfected with HaloTag fusion constructs of various RAS isoforms, as well as HVRs, in six-well plates. Transfection was conducted using Fugene (Promega) reagent and 1.5 µg DNA per well. The day after transfection, cells were transferred on to clean coverglass (#1.5, plasma-cleaned) and allowed to grow for another day. On the day of imaging, coverslips were washed with phosphate buffer saline and cells were labeled with 25pM fluorescent (JF646 or JF549) HaloTag ligand, which covalently binds to the HaloTag-RAS molecules. Fluorescent HaloTag ligands were obtained from Dr. Luke Lavis at (HHMI, Janelia Farm, Ashburn, VA). These fluorescent dyes are highly photostable and resistant to photobleaching (*Grimm et al., 2016*).

## Western blotting

MEF cells were seeded in a 6-well TC plate in 2 mL of DMEM (Gibco) media supplemented with 10% FBS (GE Healthcare Life Sciences) and 2 mM L-glutamine (Life Technologies). Cells were allowed to proliferate in a 37°C 5% $CO_2$ incubator for 24 hr until 80% confluent, and then lysed. In signal-rescue experiments, HaloTag-KRAS4b MEFs were washed twice with dPBS and then subjected to an additional 18 hr of serum starvation in 2 mL per well of 2 mM L-glutamine DMEM media with 0.1% FBS. Signaling was rescued by media aspiration, a single dPBS wash, and addition of 2 mL DMEM with 10% FBS for either 15 or 60 min. Cells were then washed once with ice cold DPBS and lysed with a cell scraper in a mixture of 20 mM Tris HCl, 150 mM NaCl, 1 mM EDTA + EGTA, 1% Triton, and Halt protease and phosphatase inhibitor cocktail (Thermo Scientific). Lysates were then centrifuged at 16,000xg for 15 min at 4°C, and the supernatant collected in a separate 1.5 mL microcentrifuge tube. Protein concentrations were analyzed using a BCA kit (Thermo Scientific), and equal amounts of protein were combined with Bolt LDS buffer (Invitrogen), Bolt Sample Reducing buffer (Invitrogen), and deionized water. Sample mixtures were heated to 100°C and mixed at 500 rpm for 5 min. 25 µg protein was loaded per well onto a Bolt 4–12% 10-well Bis-Tris gel (Invitrogen) and run at 125V in Bolt MES SDS running buffer (Invitrogen). Proteins were transferred onto a nitrocellulose membrane using Thermo Fisher Scientific's iBlot 2 Dry Blotting System and transfer stacks at 20V for 1 min, 23V for 4 min, and then 25V for 2 min. Membranes were blocked in Odyssey Blocking buffer (LI-COR) for one hour at room temperature, and then incubated at 4°C overnight in Odyssey Blocking buffer containing 0.1% Tween 20 and the following antibodies: ERK 1/2 (mouse monoclonal, Cell Signaling Technology no. 4696), pERK 1/2 (rabbit monoclonal, Cell Signaling Technology no. 4370), MEK 1/2 (mouse monoclonal, Cell Signaling Technology no. 4694), pMEK 1/2 (rabbit monoclonal, Cell Signaling Technology no. 9154), pan AKT (mouse monoclonal, Cell Signaling no. 2920), pAKT (Ser473) (rabbit monoclonal, Cell Signaling no. 4060), vinculin (mouse monoclonal, Sigma-Aldrich no. V9131), HaloTag (mouse monoclonal, Promega no. G9211), and pan RAS (mouse monoclonal, Thermo Scientific kit 16117). Membranes were washed three times in 0.05% Tween TBS for 5 min each and incubated in IRDye secondary antibodies (Goat anti-Mouse 680RD, LI-COR; Goat anti-Rabbit 800CW, LI-COR) diluted at 1:10,000 in 0.1% Tween Odyssey Blocking buffer for one hour at room temperature. Membranes were washed as described before, and images were captured using the LI-COR Odyssey CLx Imaging System.

## Single molecule microscopy

Single molecule imaging was carried out on the Nikon N-Storm microscope equipped with an APO x100 TIRF objective of 1.49NA (Nikon, Japan). A Tokai hit stage incubator (Tokai Hit Co, Ltd, Japan) was used to provide 5% $CO_2$ while maintain the temperature at 37°C for live cells. Labeled molecules (with JF646/JF549 dyes) associated with membrane were illuminated under TIRF mode. The JF549 dye was excited with the 561 nm laser which is one of the four laser lines from the Agilent laser module of the Nikon N-STORM system (*Sergé et al., 2008*), the JF646 dye was excited with the 647 nm laser line. The output laser beam was coupled into the Nikon TIRF box through a single mode fiber and focused into the back focal plane of the objective to form a parallel beam for wide field operation. The TIRF illumination was achieved by changing the illumination angle through the Nikon TIRF box controlled by the Nikon software (NIS- Elements AR 4.4). Fluorescent signals from each molecule were recorded with a thermoelectric-cooled EMCCD camera with 16 µm pixel size, (iXon Ultra DU-897, Andor Technologies, USA). Single molecule tracking was implemented by time-lapse imaging of the molecules under continuous illumination at 10 ms exposure for a total of up to 3000 frames with zero delay time between frames. At this frame rate, membrane-bound molecules appear as transient, diffraction-limited fluorescence spots. An area of $16 \times 16$ µm$^2$ of the plasma membrane in the cytoplasmic region of each cell was imaged.

## Single molecule tracking data processing

The ImageJ-based single molecule tracking plugin, TrackMate (*Tinevez et al., 2017*) or Localizer (*Dedecker et al., 2012*) was used to create tracks from the time-lapse movies. Single molecules were identified as spots from each frame of the time-lapse movies with the eight-way adjacency particle detection algorithm with 30 GLRT (*Sergé et al., 2008*) sensitivity and a PSF of 1.3 pixels. Subresolution spot accuracy was achieved using a 2D Gaussian fit function for estimating the position of

the PSF for each frame. These spots were linked into tracks given certain criteria and cut off. The single molecule spot detection and tracking parameters were kept consistent across all experiments. These tracks were organized and exported for HMM and MSD analysis using a semi-automated workflow, developed in Matlab (Mathwork, Natick, MA), on a multi-core Mac Pro or a high-performance batch cluster (ABCC, FNLCR) (*Figure 1—figure supplement 1*). Tracking data was obtained in multiple replicates for each and every condition (~8000 tracks and 16 cells). Mean and standard error of mean (S.E.M) were estimated from the results obtained from HMM and MSD analysis.

## HMM analysis with vbSPT software

Tracks from multiple cells were combined as input to the vbSPT (*Persson et al., 2013*) for HMM analysis. This analysis extracts discrete diffusive states in the single molecule trajectories and transition rates between states during heterogenous intracellular diffusion. Typically, the number of iterations and bootstrapping were set to 250. Diffusion coefficients and occupancies obtained from the analysis were compared between days to get statistical significance.

## MSD analysis

The Matlab based TrackArt (*Matysik and Kraut, 2014*) software was used to do MSD analysis. Tracking data of the same type of molecules from multiple cells were combined before input into the TrackArt. MSD analysis was based on all tracks input and the result was the average of many samples, which reflected the collective behavior of the type of molecules. The MSD vs. time plot can explicitly show molecules which are confined in motion and to what extent they are confined.

## Molecular modeling and simulations

For modeling and simulation, the wild type 19-residue HVR peptide was cut from a KRAS4b/GDP crystal structure (PDB #5TAR), and the farnesylated Cys185 was methylated for lipid insertion. The wild-type full-length KRAS4b model was built by attaching the 19-residue HVR to the KRAS4b/GMPPNP crystal structure with the GMPPNP modified into GTP. Each mutant was built by mutating the corresponding amino acid residues using MOE suite of programs (*Chemical Computing Group Inc, 2013*). Standard amino acids in HVR, full-length KRAS4b and their mutants are modeled using CHARMM36 Force Field (FF) (*Best et al., 2012*; *Klauda et al., 2010*), and the C-terminus farnesyl moiety was modeled using parameters derived by Neale and Garcia (*Neale and García, 2018*). Guanine nucleotide parameters are based on guanosine monophosphate and pyrophosphate parameters developed for use with CHARMM36 nucleic acids (*Denning et al., 2011*), while explicit solvents were modeled using TIP3P water model with CHARMM modification (*Jorgensen et al., 1983*; *MacKerell et al., 1998*). GROMACS was used for all HVR and mutant simulations (*Abraham et al., 2014*), while GPU accelerated PMEMD from AMBER16 was employed for all full-length KRAS and mutants simulations (*Case et al., 2016*).

For HVR mutant simulations, each of eight peptides were inserted into a pre-equilibrated and solvated POPC (80%) – POPS (20%) bilayer composed of 200 lipids by placing the farnesyl moiety inside the bilayer. Counterions were subsequently added to each system, resulting in a model consisting of the HVR (mutant), 160 POPC lipids, 40 POPS lipids, ~13,000 water molecules, 98 $K^+$ ions and 68 $Cl^-$ ions (for wild type). Each system was minimized to remove close contacts, followed by a 1.5 µs NTP ensemble MD simulation at 310 K and 1 bar.

For the full-length KRAS4b mutant simulations, each of eight proteins, with GTP and $Mg^{2+}$ bound, were inserted into a pre-equilibrated and solvated POPC (80%) – POPS (20%) bilayer composed of 440 lipids by placing the farnesyl moiety inside the bilayer. Counterions were subsequently added to each system, resulting in a model consisting the KRAS4b (mutant), 352 POPC lipids, 88 POPS lipids,~42,000 water molecules, 405 $K^+$ ions and 317 $Cl^-$ ions (for wild type). Each system was minimized to remove close contacts, followed by a 1.0 µs NTP ensemble MD simulations at 310 K and 1 bar.

For the POPS concentration simulations, nine lipid composition, 100% POPC, 95% POPC – 5% POPS, 90% POPC – 10% POPS, 85% POPC – 15% POPS, 80% POPC – 20% POPS, 75% POPC – 25% POPS, 70% POPS – 30% POPS, 65% POPC – 35% POPS and 60% POPC – 40% POPS, were prepared, pre-equilibrated and solvated. Next, the 19-residue wild type HVR was inserted into each lipid by placing the farnesyl moiety into the lipid. Each system consists of the wild type HVR, a total

of 200 lipids, ~13,000 water molecules and 0.15M counterions (K$^+$ and Cl$^-$). Each system was subsequently minimized followed by a 1.5 µs NTP ensemble MD simulations at 310 K and 1 bar.

## Acknowledgements

We thank Livermore Computing and the Grand Challenge Program for the computer time. A portion of this work was performed under the auspices of the U.S. Department of Energy by Lawrence Livermore National Laboratory under Contract DE-AC52-07NA27344. LLNL-JRNL-771099-.

## Additional information

### Funding

| Funder | Grant reference number | Author |
|---|---|---|
| National Cancer Institute | NIH Contract HHSN261200800001E | De Chen<br>Prabhakar R Gudla<br>John Columbus<br>Karen Worthy<br>Megan Rigby<br>Suman Mukhopadhyay<br>Katie Powell<br>William Burgan<br>Vanessa Wall<br>Dominic Esposito<br>Dhirendra Simanshu<br>Dwight V Nissley<br>Thomas Turbyville |
| U.S. Department of Energy | Joint Design of Advanced Computing Solutions for Cancer (JDACS4C) | Yue Yang<br>Felice C Lightstone |

The funders had no role in study design, data collection and interpretation, or the decision to submit the work for publication.

### Author contributions

Debanjan Goswami, Conceptualization, Data curation, Software, Formal analysis, Investigation, Visualization, Methodology; De Chen, Conceptualization, Data curation, Software, Formal analysis, Supervision, Validation, Investigation, Visualization, Methodology; Yue Yang, Conceptualization, Software, Formal analysis, Investigation, Visualization; Prabhakar R Gudla, Conceptualization, Data curation, Software, Formal analysis, Investigation, Methodology; John Columbus, William Burgan, Resources, Investigation, Writing - review and editing; Karen Worthy, Validation, Investigation, Visualization; Megan Rigby, Formal analysis, Validation, Investigation, Writing - review and editing; Madeline Wheeler, Formal analysis, Validation, Writing - review and editing; Suman Mukhopadhyay, Investigation, Writing - review and editing; Katie Powell, Resources, Validation, Investigation; Vanessa Wall, Resources, Formal analysis, Investigation; Dominic Esposito, Resources; Dhirendra K Simanshu, Conceptualization, Investigation, Visualization, Writing - review and editing; Felice C Lightstone, Conceptualization, Formal analysis, Supervision; Dwight V Nissley, Conceptualization, Supervision; Frank McCormick, Conceptualization, Methodology, Writing - review and editing; Thomas Turbyville, Conceptualization, Supervision, Methodology, Project administration

### Author ORCIDs

Debanjan Goswami (iD) https://orcid.org/0000-0001-5910-3811
Felice C Lightstone (iD) http://orcid.org/0000-0003-1465-426X
Thomas Turbyville (iD) https://orcid.org/0000-0003-2638-9520

### Decision letter and Author response

Decision letter https://doi.org/10.7554/eLife.47654.sa1
Author response https://doi.org/10.7554/eLife.47654.sa2

# Additional files

## Supplementary files

• Supplementary file 1. Statistical analysis of data. For mean-squared displacement plots, an unpaired, two-tailed *t*-test was performed on three displacement values associated with the later time points in the plots. p-Values are presented to indicate significance in the difference between each pair of conditions. For HMM analysis, diffusion co-efficient and occupancy values are compared using an unpaired, two-tailed *t*-test to find significance in the difference between pairs of interest. p-Values are presented only for those tests that have passed the significance test ($p < 0.05$).

• Supplementary file 2. Table of key resources used in the study.

• Transparent reporting form

## Data availability

Trajectories and inputsfor the molecular dynamic simulations, have been provided on the webpage at https://bbs.llnl.gov/data.html. Imaging data has been uploaded to Zenodo:http://doi.org/10.5281/zenodo.3697985.

The following dataset was generated:

| Author(s) | Year | Dataset title | Dataset URL | Database and Identifier |
|---|---|---|---|---|
| Goswami D, Chen D, Yang Y, Gudla RP, Columbus J, Worthy K, Rigby M, Wheeler M, Mukhopadhyay S, Powell K, Burgan W, Wall V, Esposito D, Simanshu D, Lightstone FC, Nissley DV, McCormick F, Turbyville T | 2020 | Membrane interactions of the globular domain and the hypervariable region of KRAS4b define its unique diffusion behavior | http://doi.org/10.5281/zenodo.3697985 | Zenodo, 10.5281/zenodo.3697985 |

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
