## [Decision Letter]

**Acceptance summary:**

This interesting study extends previous analyses of KRas4b function by investigating diffusion by single particle tracking. The authors demonstrate that KRas4b properties differ from other Ras-like molecules and that the diffusion can be assigned to three different states. It is proposed that the rapidly diffusing KRas4b corresponds to free KRas4b and that the two slowly diffusing populations correspond to KRas4b assembled into signaling complexes.

**Decision letter after peer review:**

Thank you for submitting your article "Membrane interactions of the globular domain and the hypervariable region of KRAS4b define its unique diffusion behavior" for consideration by *eLife*. Your article has been reviewed by three peer reviewers, one of whom is a member of our Board of Reviewing Editors, and the evaluation has been overseen by Philip Cole as the Senior Editor. The reviewers have opted to remain anonymous.

The reviewers have discussed the reviews with one another and the Reviewing Editor has drafted this decision to help you prepare a revised submission.

Summary:

This is an interesting study that extends previous analyses of KRas4b function by investigating diffusion by single particle tracking. The authors demonstrate that KRas4b properties differ from other Ras-like molecules and that the diffusion can be assigned to three different states. It is proposed that the rapidly diffusing KRas4b corresponds to free KRas4b and that the two slowly diffusing populations correspond to KRas4b assembled into signaling complexes. In general, the data presented support the conclusions. However, there are a number of issues that should be addressed:

Essential revisions:

1) The Discussion section largely reiterates the findings and only briefly mentions the 'new model' depicted in Figure 6C (and mentioned in the Abstract). It is difficult to see how the data in this paper provides evidence for this sequence of events. For example, in the fourth paragraph of the Discussion it is stated that the 4bHVR domain recruits and clusters negatively charged lipids (in the free diffusion state) but such an induced lipid cluster would have an impact on KRAS4b diffusion (not observed experimentally). It is also not clear why GTP loading of wt KRAS4b should only occur in the intermediate and slow diffusion states but that's need to result in the step-wised assembly process of Figure 6C. The data shown does not formally exclude that the binding of the effector proteins results in slow(er) diffusion. Indeed, Figure 6C does not distinguish between the fast and slow diffusing states of RAS4b when that is what the paper focuses on.

2) It is intriguing that the confinement behavior of KRAS4b is caused mainly by the G-domain and not the electrostatic interaction facilitated by the hypervariable region (HVR) and inversely, that the fast diffusion is only regulated by the charged residues in the HVR. Can the MD simulation give insights into the importance of the position of the lysine (i.e. 4bHVR-5EA versus 4bHVR-5EB)? Moreover, these mutational data require a comparison of the effects of these mutations on the biological signaling properties of KRas4b. Do the authors rely on published data for this comparison – if so, these data should be cited and discussed. If not, this data should be provided in this study.

3) Please confirm that data in Figure 5C is a 3-state model (obtained from the vbSPT analysis)? [Is only Figure 3 analyzed as a 2-state model?]. Please also provide details of the statistical analysis. If we understood the figure correctly, the only significant difference in Figure 5D is in the occupancy of the fast and intermediate diffusing states between 4B-Q61R and 4B-Y40C-A61R (while there is no significant difference to 4B-WT). If that is correct, we don't understand the authors' statement that the Q61R mutant has 'significantly more molecules in the slow-moving component' or the GTP-loading statement at the end of the Introduction.

4) Statistical analysis such as two-way ANOVA is required to describe the significant differences. This is a serious problem throughout the study.

5) The HeLa cell tracking data were obtained using Fugene-transfected cells. Most likely this led to highly variable levels of KRas4b expression in these cells. This is not addressed in the study. Did the level of expression affect the observed diffusion properties of KRas4b?

6) Figure 2. It was stated that "MSD analysis of single molecule trajectories in living cells showed that the 4b-ESR mutant (H-like: D126E/T127S/K128R) had less confinement (Figure 2C)" in the manuscript. However, it is unclear whether there is a significant difference. The authors described "vbSPT analysis showed that its fast-moving component increased, and its slow-moving components decreased (Figure 2D)" but no statistical analysis is shown.

7) Figure 5. It is unclear why the authors used the Y40C, Q61R double mutant to test the Y40C effect. The authors need to provide a rationale and/or test the Y40C mutant.

8) Supplementary figure 1. It was stated "Finally, we found that the levels of pAKT, pMEK and pERK are at levels comparable to WT MEF cells as measured in Western blots (Supplementary Figure 1A)." This refers to Supplementary Figure 1D. Wild type MEF and KRas fl/fl need to be included as controls.

---

## [Author Response]

Essential revisions:1) The Discussion section largely reiterates the findings and only briefly mentions the 'new model' depicted in Figure 6C (and mentioned in the Abstract). It is difficult to see how the data in this paper provides evidence for this sequence of events. For example, in the fourth paragraph of the Discussion it is stated that the 4bHVR domain recruits and clusters negatively charged lipids (in the free diffusion state) but such an induced lipid cluster would have an impact on KRAS4b diffusion (not observed experimentally). It is also not clear why GTP loading of wt KRAS4b should only occur in the intermediate and slow diffusion states but that's need to result in the step-wised assembly process of Figure 6C. The data shown does not formally exclude that the binding of the effector proteins results in slow(er) diffusion. Indeed, Figure 6C does not distinguish between the fast and slow diffusing states of RAS4b when that is what the paper focuses on.

We redrew the model in Figure 6 to more accurately reflect the data, with the following points to be made in the manuscript:

– The fast diffusing species is only dependent on the processed HVR: the positively charged lysines and the farnesyl tail are required for the observed fast diffusion state, as indicated in Figure 3. Decreasing or reversing the positive charge on the HVR increases the diffusion rate. Specifically, while we do get tethering and observable diffusion when we mutate the lysines (and either decrease or reverse the charge density on the HVR), the diffusion becomes faster as the positive charge density decreases or is reversed. We infer that the fast diffusion component in wild type RAS is dependent on lipid clustering.

– Though we do not directly observe the recruitment of negatively charged lipid molecules around the HVR in experimental data (as this is beyond our imaging capabilities), the atomistic MD simulations clearly show this to be the case. The simulation also shows (Figure 4) that mutations in the lysine residues increases the distance of the HVR from the plasma membrane (there are less protein-lipid contacts), and these data are well correlated with our single molecule tracking data with these same mutations in Halo-4bHVR constructs.

– There is a statistically significant difference in Figure 5 in the confinement of KRAS4b-Q61R (which is highly resistant to GTP hydrolysis, and therefore we can assume is fully GTP loaded) and the KRAS4b-Q61R-Y40C mutant. The Y40C mutant is unable to engage with the effector, and so, the loss of accumulation in the intermediate state is an indication that increasing confinement is correlated with signaling.

We edited the manuscript for clarity to make sure that we are addressing the reviewer’s concerns.

2) It is intriguing that the confinement behavior of KRAS4b is caused mainly by the G-domain and not the electrostatic interaction facilitated by the hypervariable region (HVR) and inversely, that the fast diffusion is only regulated by the charged residues in the HVR. Can the MD simulation give insights into the importance of the position of the lysine (i.e. 4bHVR-5EA versus 4bHVR-5EB)? Moreover, these mutational data require a comparison of the effects of these mutations on the biological signaling properties of KRas4b. Do the authors rely on published data for this comparison – if so, these data should be cited and discussed. If not, this data should be provided in this study.

We hypothesize that the 4bHVR-5EB is not able to bind to the chaperone PDE-Δ; and therefore, is not transported to the plasma membrane as efficiently (Dharmaiah et al., 2016). As seen in Figure 3B, there are few trajectories for this mutant, and in Figure 3—figure supplement 1, there is poor membrane localization. We cite the following publications ((Terrell et al., 2019; Zhou et al., 2017) showing that signaling is indeed impacted by mutations in the HVR. Our data support an interpretation that lipid sorting by the HVR is required for proper assembly with the effectors upon GTP loading. The increased density of the negatively charged lipid around RAS may recruit other RAS molecules, and even effectors. However, we do not have direct evidence of this. Future work using two-color simultaneous imaging and tracking may be able to shed light on these interactions.

3) Please confirm that data in Figure 5C is a 3-state model (obtained from the vbSPT analysis)? [Is only Figure 3 analyzed as a 2-state model?]. Please also provide details of the statistical analysis. If we understood the figure correctly, the only significant difference in Figure 5D is in the occupancy of the fast and intermediate diffusing states between 4B-Q61R and 4B-Y40C-A61R (while there is no significant difference to 4B-WT). If that is correct, we don't understand the authors' statement that the Q61R mutant has 'significantly more molecules in the slow-moving component' or the GTP-loading statement at the end of the Introduction.

Because over-fitting data is a danger when using the vbSPT HMM method, we use the following approach to test the validity of the number of states.

– We have observed a background level of immobile single molecules in our videos. We do not feel that we can filter these out and exclude them from the analysis, because they are still diffusing, albeit at a slow rate. However, based on many experiments, we believe these molecules to be artifacts of the experimental set up: perhaps molecules that are trapped between the glass and the cell membrane interface, or fluorophores that have attached non-specifically to cell structures.

– When we do vbSPT HMM analysis and observe the fractional occupancy of the slow state that includes these immobile molecules to be very low (<5%), and we observe in the corresponding MSD analysis that the plot is relatively straight and unconfined, we feel confident that these trajectories are better fit by a two-state model.

– We have gone in the other direction with our data and fit to four-state models. In this case, we see that the values of the diffusion rates reported for the slower states are too close together to be an accurate reflection of the underlying data—the models over-fit the data.

– The most salient point to be made is that KRAS4b diffusion is distinct from the other RAS isoforms, both when we look by MSD and vbSPT. We see KRAS4b molecules behaving with greater confinement, suggesting that these molecules are exploring a structurally distinct region of the cell membrane, and that these molecules are interacting with lipids and effectors in a way that is distinct from the other isoforms. We do not study the assembly process of the other isoforms; this is a topic for future work.

4) Statistical analysis such as two-way ANOVA is required to describe the significant differences. This is a serious problem throughout the study.

We conducted Student-T Tests on the data throughout the study, and appropriately labelled the figures, and clarified statements of significance in the manuscript. We provide a table for all the pairwise comparisons (Supplementary file 1: Statistics). This statistical test was chosen in order to focus on the portion of the MSD plots that shows the behavior of the longer trajectories. Specifically, it is in the longer trajectories that the behavior of confinement vs. more free diffusion is observed.

5) The HeLa cell tracking data were obtained using Fugene-transfected cells. Most likely this led to highly variable levels of KRas4b expression in these cells. This is not addressed in the study. Did the level of expression affect the observed diffusion properties of KRas4b?

We generated a DOX inducible, Halo-KRAS4b HeLa cell line to test the effect of density on diffusion. As seen in Figure 1—figure supplement 3D, increasing density of KRAS4b does not have an appreciable effect on diffusion rates, or the fractional occupancy of KRAS4b in these states.

6) Figure 2. It was stated that "MSD analysis of single molecule trajectories in living cells showed that the 4b-ESR mutant (H-like: D126E/T127S/K128R) had less confinement (Figure 2C)" in the manuscript. However, it is unclear whether there is a significant difference. The authors described "vbSPT analysis showed that its fast-moving component increased, and its slow-moving components decreased (Figure 2D)" but no statistical analysis is shown.

We see a statistical difference between 4b-GNK and KRAS4b and indicate this in the Figure 2C. The difference between 4b-ESR and KRAS4b is not statistically significant, however, it clearly shows a trend towards less constrained diffusion (one of the three experiments conducted for this mutant on different days showed unusual variability compared to the other two days. This introduced more error into the analysis, as can be seen by the error bars.)

7) Figure 5. It is unclear why the authors used the Y40C, Q61R double mutant to test the Y40C effect. The authors need to provide a rationale and/or test the Y40C mutant.

We include data for both mutants and made the changes in the text and Figure 5. We used the Q61R mutant to show the effect of GTP loading.

8) Supplementary figure 1. It was stated "Finally, we found that the levels of pAKT, pMEK and pERK are at levels comparable to WT MEF cells as measured in Western blots (Supplementary Figure 1A)." This refers to Supplementary Figure 1D. Wild type MEF and KRas fl/fl need to be included as controls.

We performed the Western as requested and include the data in Figure 1—figure supplement 4.